# Hyperoxia induces glutamine-fuelled anaplerosis in retinal Müller cells

Charandeep Singh [1], Vincent Tran[1], Leah McCollum[1], Youstina Bolok[1], Kristin Allan[1,2], Alex Yuan[1], George Hoppe[1], Henri Brunengraber[3] & Jonathan E. Sears [1,4✉]

Although supplemental oxygen is required to promote survival of severely premature infants, hyperoxia is simultaneously harmful to premature developing tissues such as in the retina. Here we report the effect of hyperoxia on central carbon metabolism in primary mouse Müller glial cells and a human Müller glia cell line (M10-M1 cells). We found decreased flux from glycolysis entering the tricarboxylic acid cycle in Müller cells accompanied by increased glutamine consumption in response to hyperoxia. In hyperoxia, anaplerotic catabolism of glutamine by Müller cells increased ammonium release two-fold. Hyperoxia induces glutamine-fueled anaplerosis that reverses basal Müller cell metabolism from production to consumption of glutamine.

[1] Ophthalmic Research, Cole Eye Institute, Cleveland Clinic, Cleveland, OH 44195, USA. [2] Molecular Medicine, Case Western Reserve School of Medicine Cleveland, Cleveland, OH 44106, USA. [3] Department of Nutrition, Case Western Reserve School of Medicine Cleveland, Cleveland, OH 44106, USA. [4] Cardiovascular and Metabolic Sciences, Cleveland Clinic, Cleveland, OH 44195, USA. ✉email: searsj@ccf.org

Premature infants require oxygen supplementation to survive, but excess oxygen causes retinovascular growth suppression that underlies the leading cause of infant blindness known as retinopathy of prematurity (ROP)[1]. We analyzed changes in intermediary metabolism during hyperoxia in human retinal endothelial cells (RECs) and human retinal Müller glia, which coexist through glutamine consumption and production, respectively[2]. Using a stable isotope labeling technique in human RECs, primary mouse Müller glial cells and a human Müller glial cell line (MIO-M1) in culture, here we show that hyperoxia decreases entry of glycolytic carbon into the tricarboxylic acid cycle (TCAC) and induces utilization of glutaminolytic carbon in Müller cells. In hyperoxia, catabolism of glutamine increased ammonium release by twofold. Hyperoxia induces glutamine-fueled anaplerosis that reverses basal Müller cell metabolism from production to consumption of glutamine.

Retinal Müller cells are linked functionally to RECs through the synthesis of glutamine[3]. Müller cells convert lactate and aspartate to glutamine via the TCAC flux[4]. Glutamine produced by Müller cells is essential to REC proliferation and migration[5,6]. REC-specific glutamine lyase (GLS) knockout mice exhibit compromised blood vessels[5]. GLS1 and GLS2 isoforms are differentially expressed in RECs; the latter is more concentrated in endothelial tip cells necessary to new blood vessel formation[7]. Glutamine synthetase (GS) ablation prevents normal development of retinal vasculature, further confirming the importance of glutamine to endothelial cell growth and development[8]. Previous studies have also reported that RECs express the glutamine transporter SLC1A5[9].

The high glycolytic rate in cancer cells and endothelial cells has been proposed to be beneficial in supporting quick production of ATP and to produce biosynthetic molecules for the serine and pentose phosphate pathways[6]. However, previous experiments in human umbilical vein endothelial cells (HUVECs) demonstrate that 90% of glucose-derived carbon is released from the cells as lactate and therefore very little amount of glucose is actually used for biomass synthesis. In contrast, measurement of glutamine consumption in HUVECs reveals that 90% of carbon from glutamine remains in endothelial cells implying its importance for biomass production[5]. This suggests that glutamine is required for proliferation and development of endothelial cells[10]. The dominant source of glutamine in the retina is the Müller cell.

The effect of hyperoxia on metabolism has mostly been linked to the loss of the mitochondrial complexes of the electron transport chain[11,12]. These studies have used indirect measures of metabolic function such as oxygen consumption or absence of the mitochondrial complex subunit protein levels[13]. Here, we present the effects of hyperoxia on intermediary metabolism using stable isotope-labeled substrates to demonstrate (1) the loss of pyruvate-derived citrate production in hyperoxic Müller cells, (2) hyperoxia-induced consumption of glutamine to feed the TCAC through anaplerosis, and (3) increased release of ammonium by hyperoxic Müller cells in culture.

## Results

**Hyperoxia inhibits glucose-derived glutamine production**. Since Müller cells are known to produce glutamine for other cell types in the retina[2], we used [$^{13}C_6$]glucose to compare synthesis of glutamine from glycolytic carbon in normoxia and hyperoxia (Fig. 1a). We first used MIO-M1, an immortalized cell line, to study the effect of hyperoxia on metabolism. MIO-M1 cells were isolated from human eye and behave like primary Müller cells[14]. MIO-M1 cells were first cultured in normoxia with [$^{13}C_6$]glucose media for 24 h to establish an isotopic steady state followed by either normoxia or hyperoxia for 8 h (Fig. 1b) or 24 h (Fig. 1d).

We first ensured that the isotopic steady state was established for at least 20 h before treating part of the cells with hyperoxia. Isotopic steady state here means that the isotopic enrichment for the metabolites have reached stable labeling with little or no change after 24 h and stays consistent over the next 24 h of measurement in normoxic condition (Supplementary Figs. 1 and 2). Any changes reported in isotopic steady state should reflect relative increase or decrease in the fluxes in the pathways where these metabolites are located. The percentage decrease or increase in labeling has been calculated as difference in enrichment. The isotopic steady-state pilot experiments were performed only with [$^{13}C_6$]glucose. Metabolites of glycolysis and TCAC were in isotopic steady state post 23 h after adding the labeled glucose (Supplementary Fig. 1).

After 8 h, hyperoxia caused slight but statistically significant decrease in isotopic enrichments of M3 lactate and M3 pyruvate by 7% and 8%, respectively, while M2 citrate and M2 glutamate were decreased by 75% and 80%, respectively (Fig. 1b). This shows that hyperoxia only moderately suppresses glycolysis, but drastically decreases the entry of glycolytic carbon into the TCAC. We also compared total glutamate amounts in hyperoxia vs. normoxia by summing all mass isotopomers of glutamate. There was 32% decrease in total glutamate (sum of the areas of all isotopomers) in hyperoxic condition as compared with normoxic condition (Fig. 1c).

A longer exposure to hyperoxia (from 24 h to 48 h) resulted in similar M3 lactate and M3 pyruvate derived from [$^{13}C_6$] glucose, i.e., 1.5% and 4% increase, respectively, which were not statistically significant from the 8 h time point (Fig. 1d). In contrast, M2 citrate enrichment fell even further, down to 30% after 24-h of hyperoxia as compared with normoxia (Fig. 1d). M2 glutamate was completely lost from MIO-M1 cells in hyperoxic condition (Fig. 1d) confirming the oxygen-induced decreased pyruvate entry into the TCAC and the complete block of glutamate formation from glycolytic carbon. Mass isotopomer distributions (MIDs) of citrate and glutamate highlight the relationship between glycolytic carbon entry into the TCAC and glutamate synthesis in normoxia compared with hyperoxia (Fig. 1e).

**Hyperoxia induces glutamine consumption in Müller cells**. Our [$^{13}C_6$]glucose labeling experiment indicated that MIO-M1 cells were unable to produce glutamate from glycolytic carbon in hyperoxia. However, their total glutamate levels remained only 30% lower than in normoxia suggesting that hyperoxic MIO-M1 cells derive their glutamate from an alternative source. We used [$^{13}C_5$]glutamine (Fig. 1f) to test if glutamine to glutamate conversion is increased in hyperoxia. M5 glutamate production from [$^{13}C_5$]glutamine was 14% higher in hyperoxia (Fig. 1g). Glutamine-derived M3 lactate and M3 pyruvate labeling decreased in hyperoxia, as did M2 and M4/M5 citrate (Fig. 1g). The M3 labeling of lactate and pyruvate are not related to glycolytic flux, but rather to isotopic exchanges via malic enzymes.

The TCAC metabolites produced from [$^{13}C_5$]glutamine are partly M4 labeled. We, therefore, looked at the M4 labeling of fumarate and aspartate, and found increased labeling of M4 fumarate and M4 aspartate, both by 11%, (Fig. 1g), further confirming the relatively accelerated rate of glutaminolysis in hyperoxia. The overall increase in M5 glutamate (Fig. 1h) confirmed increased relative contribution of glutamine deamidation to the glutamate pool in hyperoxia.

**Oxidative vs. reductive carboxylation in MIO-M1**. In order to determine whether variations in labeling from M5 glutamine of TCAC intermediates in hyperoxia originated by oxidative

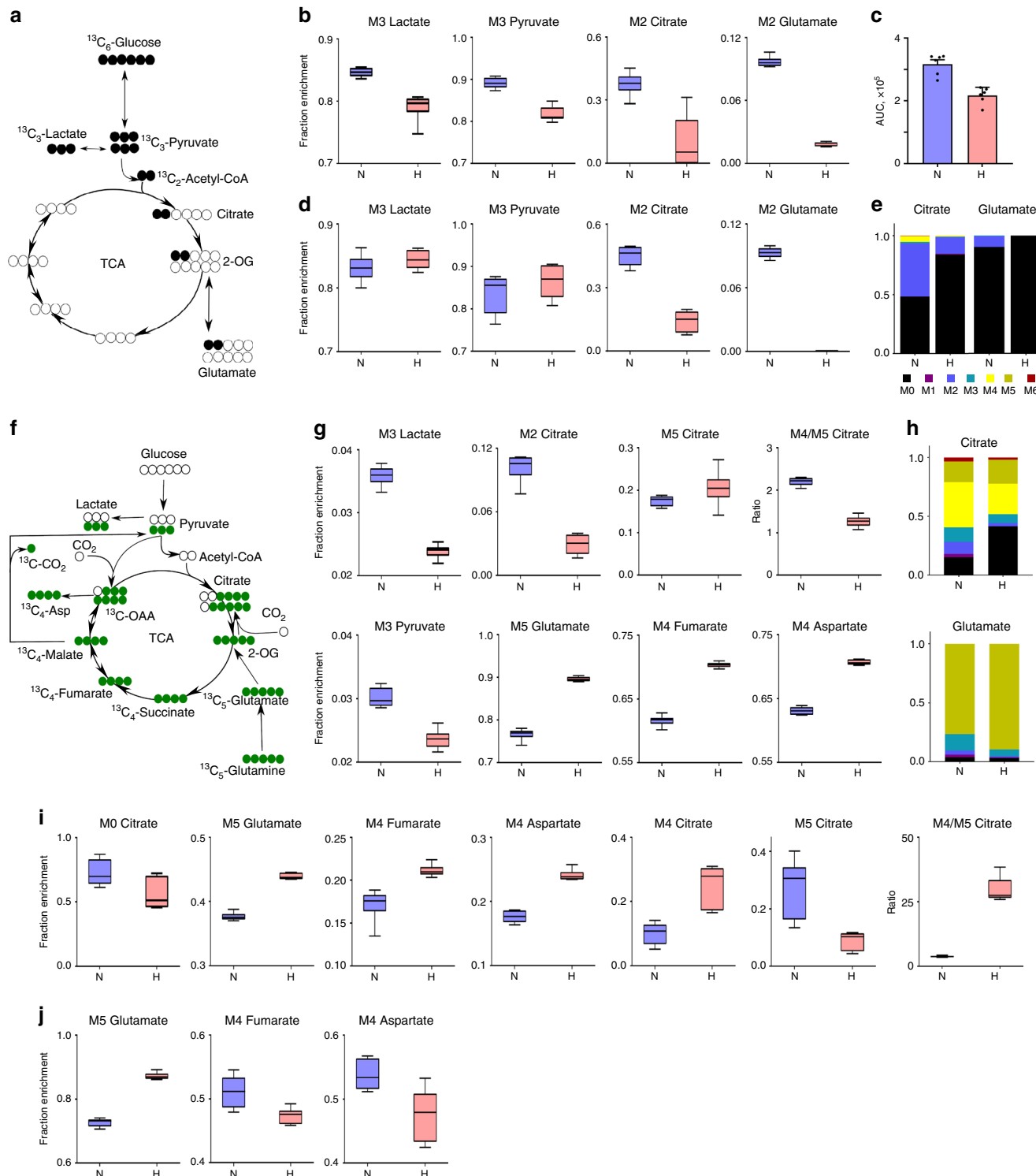

decarboxylation or reductive carboxylation, we also measured M4 vs. M5 citrate labeled from [$^{13}C_5$]glutamine. Citrate can be produced from glutamate by reductive carboxylation, which converts glutamate to citrate by back flux via isocitrate dehydrogenase or by directly converting glutamate to α-keto-glutarate (α-KG) to succinyl-CoA by oxidative decarboxylation. The former reaction yields M5 citrate, whereas the latter proceeds through the forward reactions of the entire TCAC and hence produces M4 citrate (Fig. 1f). We found that the M4/M5 ratio of citrate was decreased in hyperoxic conditions; yet overall M5 citrate trended higher without statistical significance between normoxia and hyperoxia

(Fig. 1g, h). Glutamine derived, M4 labeled TCAC metabolites such as fumarate and aspartate (Fig. 1g) were increased in hyperoxia compared with normoxia. At steady state, these findings imply that hyperoxia induces oxidative decarboxylation of α-KG yet the resulting oxaloacetate does not increase M4 citrate. Although the decrease of M4 citrate and increase in M4 fumarate confirms that α-KG is oxidatively decarboxylated, concurrently, hyperoxia might induce a change in malic enzyme flux (Fig. 1f). To get a reflection of the malic enzyme flux, we measured M3 labeled lactate and pyruvate formed by [$^{13}C_5$]glutamine and found that hyperoxia decreased M3 pyruvate and M3 lactate, implying

**Fig. 1 [$^{13}C_6$]Glucose and [$^{13}C_5$]glutamine labeling of MIO-M1 and primary Müller cells.** [$^{13}C_6$]Glucose and [$^{13}C_5$]glutamine labeling of MIO-M1 and primary Müller cells demonstrates hyperoxia-induced decreased flux from pyruvate to citrate and glucose to glutamate, but increased glutaminolytic flux into TCAC via oxidative decarboxylation, and decreased malic enzyme flux. MIO-M1 cells were cultivated in [$^{13}C_6$]glucose media for 24 h to reach isotopic steady state, then incubated in normoxia (21% $O_2$) or hyperoxia (75% $O_2$) for 8 or 24 h. **a** Schema of first round labeling of [$^{13}C_6$]glucose carbon through glycolysis and TCAC. **b** Fractional enrichment of $^{13}C$-labeled metabolites after 8 h hyperoxic treatment ($n = 6$, t-test p values: M3 lactate = 0.0001; M3 pyruvate < 0.0001; M2 citrate = 0.0006; M2 glutamate p < 0.0001). **c** Total (sum all MIDs) glutamate in normoxic vs. hyperoxic cells, after 8 h hyperoxic treatment ($n = 6$, t-test, mean ± SEM, p values = 0.0002). **d** Fractional enrichment of $^{13}C$-labeled metabolites after 24 h of hyperoxic treatment ($n = 6$, t-test p values: M3 lactate = 0.2365; M3 pyruvate = 0.2862, M2 citrate < 0.0001, M5 glutamate < 0.0001). **e** Mass isotopomer distributions of citrate and glutamate between normoxia and hyperoxia. Mass isotopomer distributions were corrected for natural isotope abundances for data represented in this figure and subsequent figures. **f** Schema of [$^{13}C_5$]glutamine carbon atoms transition through TCAC, malic enzyme, pyruvate carboxylase, and glycolytic pyruvate entry into TCAC. MIO-M1 or primary Müller cells were cultured in [$^{13}C_5$]glutamine media for 24 h, then incubated further in normoxia (21% $O_2$) or hyperoxia (75% $O_2$) for 24 h. **g** Fractional enrichment of $^{13}C$-labeled metabolites after 24 h hyperoxic treatment ($n = 6$, t-test p values: M3 lactate < 0.0001; M2 citrate < 0.0001; M5 citrate < 0.1198; M4/M5 citrate < 0.0001; M3 pyruvate < 0.0001; M5 glutamate < 0.0001; M4 fumarate < 0.0001; M4 aspartate < 0.0001). **h** Comparison of mass isotopomer distributions of citrate and glutamate between normoxia and hyperoxia. **i** Fractional enrichment of $^{13}C$-labeled metabolites in primary Müller cells after 24 h hyperoxic treatment ($n = 6$ per condition; t-test p values: M0 citrate < 0.027; M5 glutamate < 0.0001; M4 fumarate < 0.0007; M4 aspartate < 0.0001; M4 citrate = 0.0005; M5 citrate = 0.0016; M4/M5 citrate < 0.0001). **j** Fractional enrichment of $^{13}C$-labeled metabolites in primary astrocytes after 24 h hyperoxia. N normoxia, H hyperoxia, AUC area under curve. Box plots extend from 25 to 75th percentiles. Middle box line = median; whiskers represent minimal/maximal values for Fig. 2 and all subsequent box plots in Figs. 2 and 3. p values = two-sided unpaired t-test.

---

higher malic enzyme relative flux in the normoxic condition compared with hyperoxic conditions (Fig. 1g). In addition, as expected, M2 citrate was higher in normoxic conditions, which can further enhance the M4 citrate produced from M2 acetyl-CoA and M2 oxaloacetate (Fig. 1g).

**Hyperoxia induces glutamine consumption in primary Müller cells.** To determine whether oxygen-induced glutamine-fueled anaplerosis described above in MIO-M1 cells also occurs in primary Müller cells, we isolated primary Müller cells from P11 (postnatal day 11) mice. ROP is caused by oxygen supplementation necessary to resuscitating severely premature infants that unfortunately creates retinovascular growth attenuation and vasoobliteration that is the hallmark of phase 1, which subsequently leads to profound ischemia and abnormal angiogenesis in phase 2. Therefore, we chose specifically the hyperoxic phase 1 in the experimental correlate of ROP, the murine oxygen-induced retinopathy model (OIR)[15], to test both primary Müller cells and retinal explants. Hyperoxic phase 1 in the mouse model of OIR is from P7 to P12, and to be consistent with the model, we have only used cells or retinal explants from mice within the phase 1 of the model. Lysates of MIO-M1 cells, primary Müller cells, retinal explants, and primary human astrocyte cultures were compared by western blotting the levels of GS and cellular retinaldehyde binding protein (CRALBP) to ensure that cultured Müller cells and astrocytes were differentiated glia (Supplementary Fig. 3A, B). Primary Müller cells expressed similar ratios of CRALBP/GS as was found in glia from retinal explants. Primary Müller cells were cultured in 12-well plates and then incubated in the media containing [$^{13}C_5$]glutamine for 24 h to establish isotopic steady state. After 24 h, the cells were incubated either in normoxic or hyperoxic incubator for another 24 h. Intracellular metabolites were extracted and analyzed by GCMS. Like in MIO-M1 cells, we saw a similar reduction in the proportion of M4 citrate in hyperoxia, implying reduction in flux entering from glycolysis to the TCAC (Fig. 1i). In addition, we also found increased M5 glutamate, M4 fumarate, and M4 aspartate consistent with oxygen-induced increased glutaminolytic flux (Fig. 1i). These findings corroborate well our findings in MIO-M1 cells. Primary Müller cells differ from MIO-M1 cells in that they have a higher ratio of M4/M5 citrate in hyperoxia (Fig. 1i).

**Primary astrocytes increase glutamine catabolism in hyperoxia.** To further determine whether glutamine-fueled anaplerosis might

occur in other types of glia within the central nervous system, we used [$^{13}C_5$]glutamine to study the effect of hyperoxia on glutaminolytic flux in primary cortical astrocytes. Cells were again cultured in six-well plates and then incubated in [$^{13}C_5$]glutamine for 24 h to establish isotopic steady state, after which cells were incubated into normoxic or hyperoxic incubators for another 24 h. As with cultured human MIO-M1 and primary mouse Müller cells, M5 glutamate was statistically significantly higher in hyperoxic condition, implying higher rate of glutaminolysis in hyperoxia (Fig. 1j). However, primary astrocytes exhibit an interesting difference in the accumulation of metabolites downstream of αKG. In contrast to all Müller cell lines, M4 aspartate and M4 fumarate were lower in hyperoxic condition (Fig. 1j). This observation can be explained by the fact that astrocytes but not Müller cells express the AGC1 transporter protein, which allows aspartate and glutamate exchange between mitochondria and cytosol[4]. The difference in M4 aspartate and M4 fumarate enrichments in response to hyperoxia also might be due to decrease in partial isotopic dilution by cytosolic aspartate derived from proteolysis.

**Glycolytic flux in RECs is unchanged in hyperoxia.** In order to compare and contrast the effect of hyperoxia on RECs to MIO-M1 or primary Müller cells, we repeated the same labeling experiments in RECs as were undertaken in MIO-M1 and/or primary Müller cells above. Using [$^{13}C_6$]glucose (Fig. 2a), RECs demonstrated almost no change in M2 citrate (Fig. 2b), implying no decrease in the contribution of M6 glucose to citrate in hyperoxia, a distinct difference when compared with MIO-M1 or primary Müller cells treated identically. Unlike MIO-M1 or primary Müller cells, M3 lactate, and M3 pyruvate enrichments were increased in hyperoxia by 5% and 8%, respectively, while M2 glutamate enrichment was significantly decreased by 41% in hyperoxic conditions (Fig. 2b). Overall, [$^{13}C_6$]glucose labeling of RECs indicated very little change in glycolytic flux entry into the TCAC, however, glutamate production was decreased. MIDs for lactate, citrate, and glutamate are provided in Fig. 2c.

**Glutamine utilization in RECs also increases in hyperoxia.** We next measured labeling of intermediates from M5 glutamine in RECs incubated in normoxia and hyperoxia (Fig. 2d). M5 glutamate enrichment from glutaminolysis was increased in hyperoxia by 7%; M4 fumarate was increased by 4% suggesting

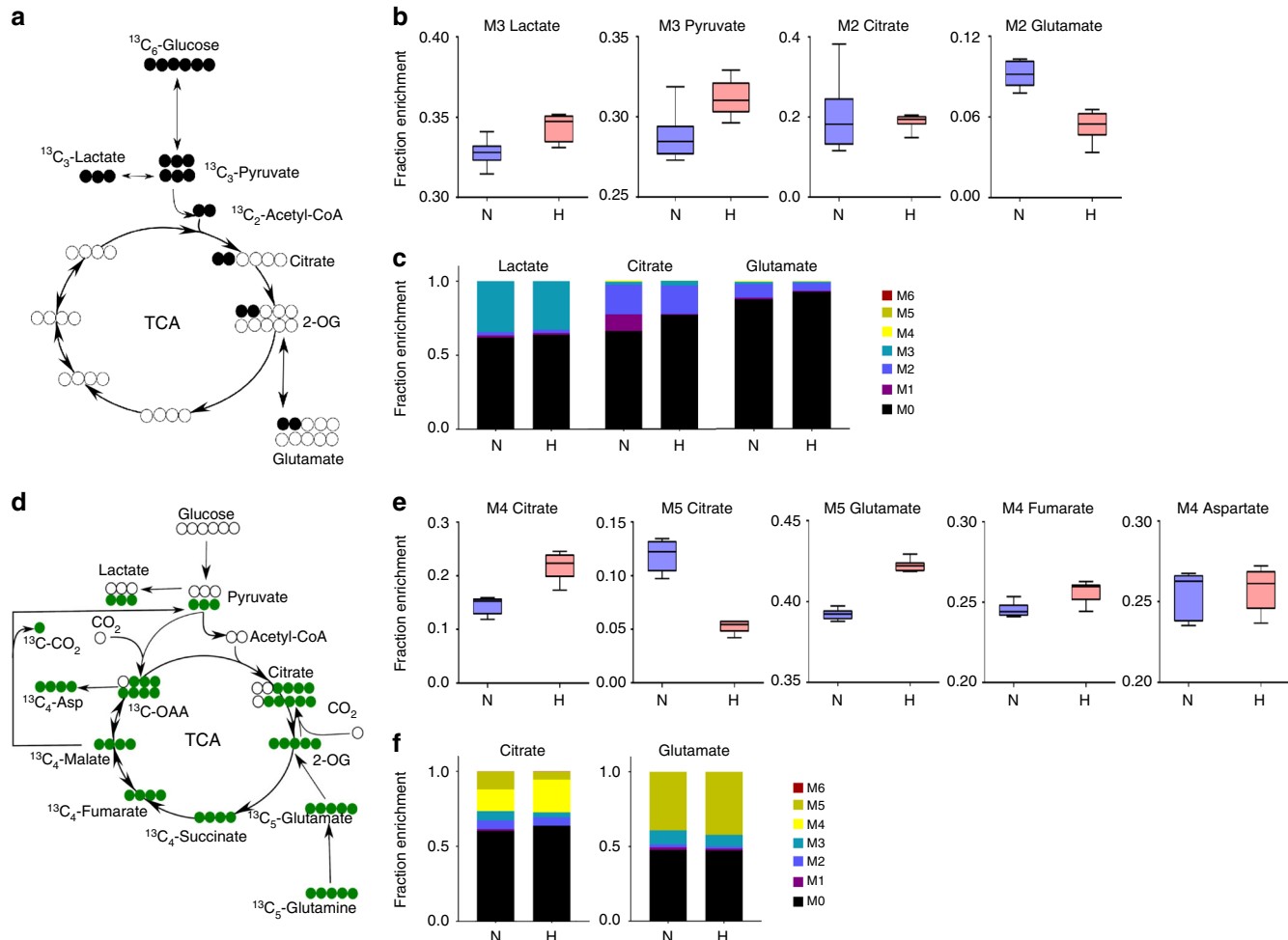

**Fig. 2 [$^{13}C_6$]Glucose and [$^{13}C_5$]glutamine labeling of retinal endothelial cells in culture.** In response to hyperoxia, [$^{13}C_6$]Glucose and [$^{13}C_5$]glutamine labeling of retinal endothelial cells in culture demonstrates no change in pyruvate to citrate flux, (a net decrease in glutamate production from glycolytic carbon, increased glutaminolytic flux feeding into TCAC via oxidative decarboxylation route, and decreased malic enzyme flux. Retinal endothelial cells were cultivated in [$^{13}C_6$]glucose containing media for 24 h to reach isotopic steady state, following which they were either incubated further in normoxia (21% $O_2$) or hyperoxia (75% $O_2$) for 24 h. **a** Schematic of the first round of [$^{13}C_6$]glucose carbon atom transition through glycolysis and TCAC. **b** Fractional enrichment of $^{13}C$-labeled metabolites after 24 h of hyperoxic treatment ($n = 6$, $t$-test $p$ values: M3 lactate = 0.0086; M3 pyruvate = 0.0138; M2 citrate = 0.7974; M2 glutamate < 0.0001). **c** Comparison of mass isotopomer distributions of lactate, citrate and glutamate between normoxia and hyperoxia. **d** REC cells were cultivated in [$^{13}C_5$]glutamine containing media for 24 h to reach isotopic steady state, following which they were either incubated further in normoxia (21% $O_2$) or hyperoxia (75% $O_2$) for 24 h. **e** Fractional enrichment of $^{13}C$-labeled metabolites after 24 h of hyperoxic treatment ($n = 6$, $t$-test $p$ values: M4 citrate = 0.0002; M5 citrate < 0.0001; M5 glutamate < 0.0001; M4 fumarate = 0.0070; M4 aspartate = 0.7713). **f** Comparison of mass isotopomer distributions of citrate and glutamate between normoxia and hyperoxia. N normoxia, H hyperoxia.

increased deamidation of glutamine and subsequent entry of glutamate into the TCAC but in contrast to Müller cells, M4 aspartate and M4 fumarate were unchanged (Fig. 2e). Furthermore, the changes in citrate labeling (M4, via oxidative decarboxylation vs. M5, via reductive carboxylation) demonstrated that hyperoxia inhibits reductive carboxylation in RECs (Fig. 2f). Glutamate labeling of REC cells clearly demonstrated increased utilization of glutamine in hyperoxia to produce TCAC compounds as evident from increased production of M5 glutamate and M4 citrate from glutamine. When examining label channeling through malic enzyme in RECs, there was little back flux of label from glutamine into pyruvate and lactate.

**Quantitative comparison of metabolites in MIO-M1 and RECs.** To understand the importance of these differences in metabolic fluxes between MIO-M1 and RECs, in normoxia and hyperoxia, we quantified the total amount of metabolites ([sum of all mass isotopomer areas of individual metabolites]/[area of M internal

standard]) in incubations of MIO-M1 and RECs. Glucose and glutamine levels were almost equal, implying that both the cell lines had equal availability of these carbon sources (Fig. 3a, b). However, the relative lactate/pyruvate ratio, which increases in aerobic glycolysis, was higher in RECs as compared with MIO-M1 cells (Fig. 3c). In addition, relative fumarate and aspartate levels were lower in RECs as compared with MIO-M1 cells, implying lower TCAC flux (Fig. 3e, f). Glutamate levels overall were reduced in MIO-M1 cells in hyperoxia (Fig. 3g).

In MIO-M1 cells, hyperoxia decreases pyruvate to citrate conversion, but increases glutamine derived fumarate and aspartate. Given that conversion of 1 mole of glutamine to α-KG releases 1 or 2 moles of ammonium, we next measured ammonium from MIO-M1 in the same cells cultured in hyperoxia for 48 h. Ammonium production, normalized to cell number, increased twofold in response to hyperoxia (Fig. 3h). Ammonium could also result from degradation of amino acids in the media. To test for this possibility, we compared ammonium

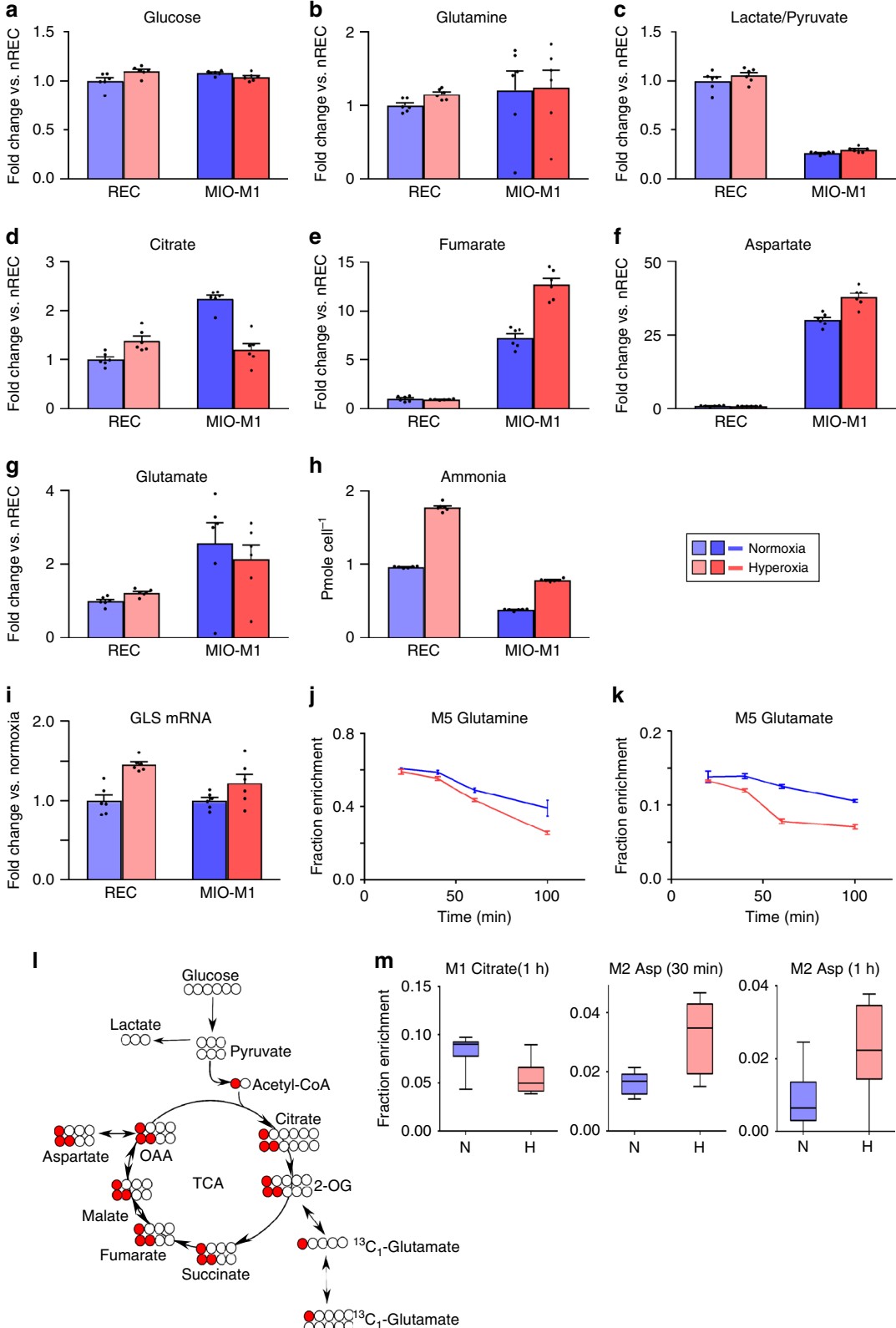

levels in spent vs. unspent media. The ammonium concentration in unspent medium was negligble (Supplementary Fig. 4A–C).

**Hyperoxia upregulates GLS and glutamine catabolism.** We next measured the expression levels of glutaminase, which has two isoforms GLS1 and GLS2. In our culture conditions, MIO-M1

cells expressed GLS1; GLS2 expression was not detectible. GLS1 expression was slightly upregulated by hyperoxia in both MIO-M1 cells and RECs (Fig. 3i). RECs also expressed higher levels of GLS1 in response to hyperoxia (Fig. 3i), whereas GLS2 expression was not detectable in RECs.

To test the effect of hyperoxia on glutamine uptake, we performed a pulse-chase experiment. P11 retinas were incubated

**Fig. 3 Total metabolite levels of retinal endothelial cells and MIO-M1 cells; retinal explants incubated with M5 glutamine or M1 acetate.**
**a–i** Comparison of total metabolite levels between retinal endothelial cells vs. MIO-M1 cells, in normoxia vs. hyperoxia; evidence of higher aerobic glycolysis in retinal endothelial cells as compared with MIO-M1 cells. **j, k** Retinal explants incubated with M5 glutamine. **l, m** Retinal explants incubated with M1 acetate. **a–i** Metabolites were extracted from confluent cells incubated with M5 glutamine, spiked with M5 ribitol internal standard, extracted and assayed by GC-MS. The sum of all MIDs were normalized to M5 ribitol. Data are presented as histograms with SEM ($N = 6$). **a** Glucose consumption, **b** glutamine, **c** lactate to pyruvate ratio, **d** citrate, **e** fumarate, **f** aspartate, **g** glutamate, and **h** ammonium, levels in RECs vs. MIO-M1 cells. **i** Retinal endothelial cell GLS1 expression is increased ($n = 6$ per condition, $p$ value 0.0002, two-sided unpaired $t$-test). GLS1 expression is also increased in MIO-M1 cells ($n = 6$ per condition, $p$ value 0.09, two-sided unpaired $t$-test). (Panels **j–k**) Pulse-chase experiment on retinal explants incubated with $[^{13}C_5]$ glutamine. Retinal explants were incubated in 5 mM $[^{13}C_5]$glutamine for 5 min and media was changed with unlabeled 5 mM glucose and 5 mM lactate containing media. **j** M5 glutamine was consumed faster in hyperoxic condition ($n = 3$ per time point per condition, mean ± SEM). The data shows that higher glutamine utilization rate in retinal explants in hyperoxia compared to normoxia. **k** M5 glutamate had a similar trend as glutamine with higher rate of utilization ($n = 3$ per time point per condition, mean ± SEM). **l** Schematic of the two rounds of $[^{13}C_1]$acetate carbon atom transition through the TCAC. **m** Fractional enrichment of $^{13}$C-labeled citrate and aspartate 30 min or 1 h after incubation in 1-$^{13}$C acetate containing media ($n = 6$ for M1 citrate (1 h) and M1 aspartate (1h); $n = 5$ for M2 aspartate (30 min) per condition). L/P ratio lactate/pyruvate ratio, nREC normoxic retinal endothelial cells, REC retinal endothelial cells, MIO-M1 Immortalized Müller cell line.

ex vivo in KRB containing 5 mM $[^{13}C_5]$glutamine for 5 min (Fig. 3j, k). P11 retinas were used because this postnatal day coincides with the hyperoxic phase 1 of the OIR model (Postnatal days 7–12). Following 5 min incubation in media containing 5 mM M5 glutamine, the media was aspirated; retinas were washed with PBS, then incubated in KRB containing 5 mM unlabeled glucose and 5 mM lactate.

Following media change, the M5 enrichment of glutamine and glutamate slowly decreased, presumably by formation of unlabeled glutamine from unlabeled glucose and lactate. These dilution rates increased in hyperoxic conditions (Fig. 3j, k).

Others have reported that acetate is metabolised in the retina in the same way as in the brain[16–21]. We used [1-$^{13}$C] acetate to test the effect of hyperoxia on the labeling of citrate by Müller cells in cultured retinal explants. Retina from P10 mice (i.e. during the hyperoxic phase 1 of the OIR model) were dissected and explants cultured in DMEM containing 5 mM glucose, 2 mM [1-$^{13}$C] acetate and 1 mM glutamine (Fig. 3l). The explants were exposed to normoxia or hyperoxia for 30 min or 1 h, before extraction of metabolites. As in isolated MIO-M1 cell culture experiments (Fig. 1b, d, g), the M1 enrichment of citrate in retinal explants was decreased in response to hyperoxia at 30 min and 1 h (Fig. 3m). Thus hyperoxia decreases the contributions of glucose, glutamine and acetate to the citric acid cycle in retina.

**No change in hyperoxia mitochondrial number or phospho-PDH.** It was reported that hypoxia leads to mitochondrial autophagy[22]. We measured concentrations of COX-IV protein, which is a marker for mitochondria, along with β-actin loading control by western blotting. We found no difference in the amount of COX-IV (Supplementary Fig. 5A), suggesting that the mitochondrial number does not change in response to 24 h of hyperoxia.

Pyruvate entry into the TCAC is regulated by pyruvate dehydrogenase kinase (PDK) protein, which inactivates pyruvate dehydrogenase (PDH) through phosphorylation. PDK expression is upregulated by HIF1α. Using western blot, PDH levels were measured in the MIO-M1 cells using β-actin as an internal control. PDH protein levels were not altered by hyperoxia (Supplementary Fig. 5B). We then measured the amount of phosphorylated PDH in the same samples, along with cells treated with HIF1α stabilizer FG-4592 as a positive control. Phospho-PDH levels were higher in HIF1α stabilized condition, but were same in normoxic and hyperoxic samples (Supplementary Fig. 5C, D). Therefore, the decrease in citrate formation in hyperoxia (Fig. 1b, d, g) is not

secondary to decreased levels of PDH or phosphorylation of PDH.

Immunohistochemistry of mitochondria (Supplementary Fig. 6A, B) demonstrates a difference in mitochondrial morphology that occurs in MIO-M1 cells in hyperoxia, changing the staining pattern without affecting overall mitochondrial density. In contrast, no significant difference was observed in hyperoxic vs. normoxic RECs (Supplementary Fig. 6C, D). Quantification of mitochondrial morphology in both RECs and MIO-M1 cells is provided in Supplementary Fig. 6E–H. It demonstrates equal density of mitochondria but definite change in morphology due to hyperoxia in MIO-M1 cells and RECs. The latter have clumped mitochondria in hyperoxia reflecting smaller fragmented and reduced branching of the mitochondrial networks. Pyruvate kinase, required for the conversion of phosphoenolpyruvate to pyruvate, has two isoforms present in the retina, PKM1 and PKM2. PKM1 and PKM2 are present in photoreceptors, whereas there is weak expression of PKM2 in the other layers of the retina[4]. PKM1 is expressed in MIO-M1 cells in culture, and its expression decreases in response to 24 h of exposure to hyperoxia (Supplementary Fig. 7).

**Discussion**
Our data obtained using both mouse Müller glial cells and MIO-M1 (an immortalized human Müller glia-like cell line), demonstrate that Müller cells in hyperoxia have a higher flux through the glutaminolytic branch to the TCAC rather than from glycolysis (Fig. 1b, d, g). The reversal of glutamine production in Müller cells in normoxia to consumption in hyperoxia has profound implications for the mechanism of oxygen-induced retinovascular growth attenuation, the causation of ROP. First, energy-dependent production of glutamine is necessary for endothelial cell survival and proliferation. Second, glutamine synthesis detoxifies ammonium, in lung epithelia, in pericentral hepatocytes, and in brain[2,23,24]. Third, neurons rely on glutamine produced by Müller cells to make glutamate for synaptic transmission[25]. High oxygen saturations therefore may interrupt the basic functions of glia critical to endothelial cell metabolism and neuronal homeostasis. The oxygen-induced release of ammonium and inhibition of glutamine synthesis could contribute to the phenomenon of hyperammonemia in low birth weight/premature infants treated with oxygen supplementation[26,27]. The schema in Fig. 4 demonstrates how oxygen excess affects the overall flow of carbon and nitrogen in both Müller cells and RECs.

The metabolic consequences of hyperoxia are simply not the opposite of hypoxia. HIF blocks entry of pyruvate into the TCAC[28]. Given that HIF concentration is decreased in hyperoxia,

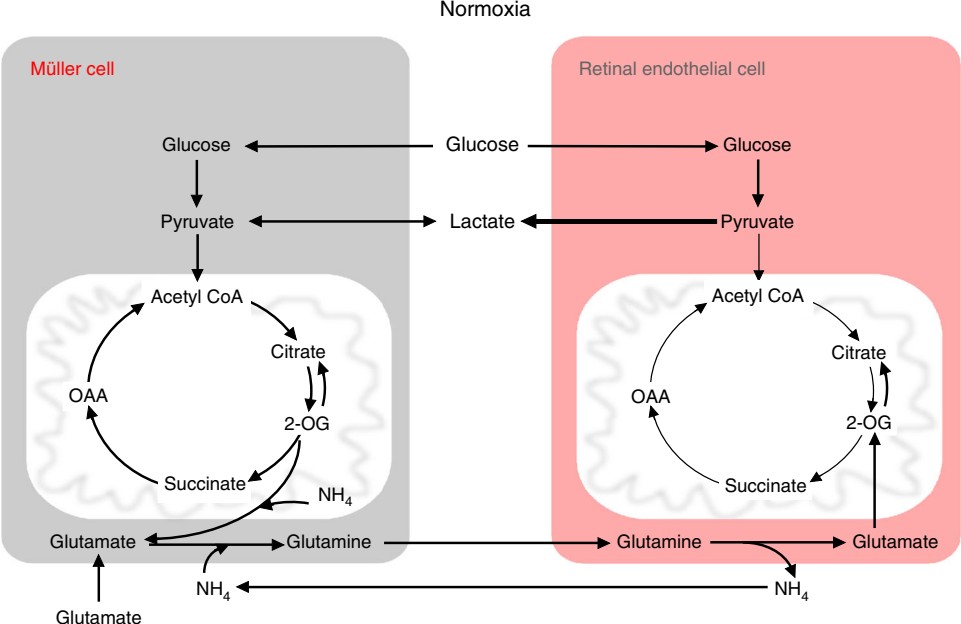

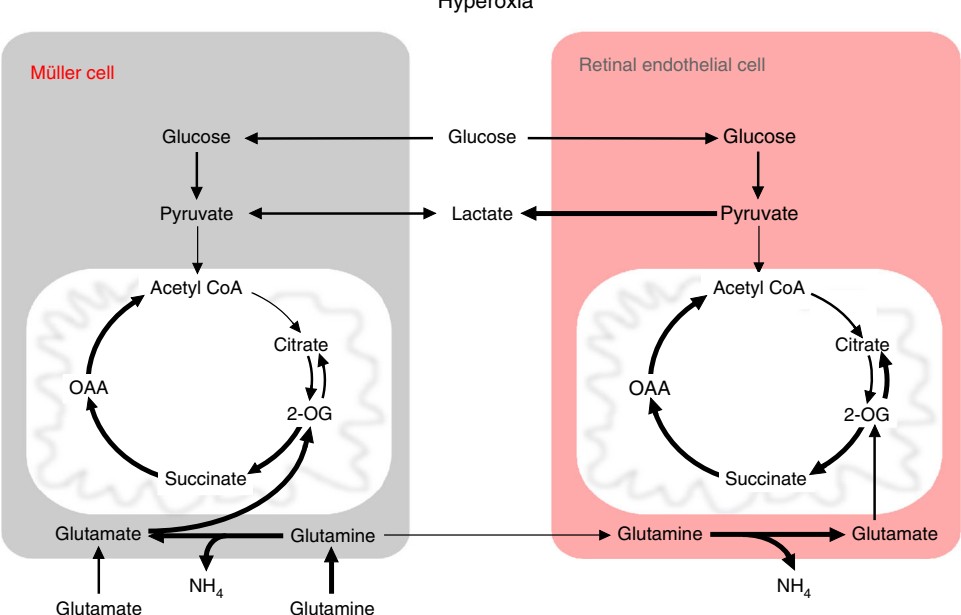

**Fig. 4 Model depicting the re-programming of metabolism of Müller and retinal endothelial cells in response to hyperoxia.** In normal conditions, Müller cells remove ammonium produced by other cell types in two steps by converting α-KG to glutamate and then glutamate to glutamine. Müller cells are the only cell type in retina known to fulfill all the glutamine requirements of the retina and are the only cells known to remove toxic ammonium formed by other cells in the retina. Retinal endothelial cells use aerobic glycolysis to produce energy and glutamine for growth. The lower panel shows the hyperoxic effect on metabolism of these cell types. In hyperoxia Müller cells stop producing glutamine and utilize it at increased rates for their energy needs by oxidative decarboxylation. Müller cells fulfill their energy needs by anaplerosis from glutamine when glycolytic flux entry to TCAC is blocked in hyperoxic conditions. This can lead to low glutamine levels in the retina for other cells types, including retinal endothelial cells. Glutaminolysis in retinal endothelial cells is also increased in response to hyperoxia, whereas, IDH back flux of citrate is decreased in response to hyperoxia. Overall hyperglutaminolysis in response to hyperoxia can lead to accumulation of toxic ammonium as well as a glutamine deficit. This would cause an overall metabolic imbalance leading to oxygen-induced growth suppression.

one would hypothesize that the flux through PDH should increase. This is not the case as shown in a previous report in lung epithelium[29]. We measured the PDH and phospho-PDH levels in hyperoxia vs. normoxia cells, and found no difference. There are four paralogs of PDK in humans, PDK1, PDK 2, PDK 3 and PDK4[30]. Phosphorylation of PDK1 is under HIF regulation via 12 phosphorylation sites. Out of those, the most common one

studied in HIF stabilized conditions is p-Ser232[31]. We specifically tested for this phosphorylated form and found no difference in hyperoxic vs. normoxic condition. Therefore, in Müller cells, there may be a mechanism other than phosphorylation of PDH controlling the entry of pyruvate into the TCAC. It is also possible that a different phosphorylation site on PDH is responsible for the PDH flux blockage in hyperoxic conditions.

There are other cells that show high rates of glutamine catabolism. For example, cancer cells oxidize glutamine to α-KG; if a monoalleleic mutation in isocitrate dehydrogenase I is present, α-KG is further reduced to the oncometabolite 2-hydroxyglutarate[32,33]. Glutamine catabolism in hypoxia is regulated by multiple pathways, such as HIF1α and c-Myc; of these, hypoxia and glucose deprivation is considered to down-regulate c-Myc expression[34–37]. In conclusion, we demonstrate that hyperoxia stimulates glutamine-fueled anaplerosis in retinal Müller cells. Such glutamine deprivation may perturb the growth and functions of retinal endothelial cells.

## Methods

**Ethics Approval Statement**. All treatments of animals (mice) were approved by the Cleveland Clinic Institutional Animal Care and Use Committee (IACUC) under study protocol number 2019-2183.

**Chemicals**. All the chemicals were purchased from Sigma-Aldrich (St. Louis, MO, USA) unless otherwise stated. The derivatization reagent MSTFA+1%TMCS was purchased from Thermo-scientific (Bellefonte, PA, USA). Stable isotope-labeled compounds were purchased from Cambridge Isotope Laboratories (Andover, MA, USA) and were reported as 99% pure. LC-MS grade methanol (HiPerSolv Chromanorm; BDH VWR International, Radnor, PA, USA) and chloroform (LiChrosolv; MilliporeSigma, Burlington, MA, USA) were used for all the metabolite extractions. DMEM media was purchased from Cleveland Clinic Media Lab. Endothelial cell media, Complete Classic Medium with Serum and CultureBoost™, was purchased from Cell Systems (Kirkland, WA, USA). Iodine was purchased from Acros organics (Bellefonte, PA, USA). CyQuant NF kit was purchased from Invitrogen (Bellefonte, PA, USA).

**Cell culture**. RECs were procured from Cell Systems (Kirkland, WA, USA). The immortalized Müller cell line MIO-M1 was a kind gift from Dr. Limb[38]. DMEM (without glucose, glutamine and pyruvate) was supplemented with 15mM glucose, 5 mM glutamine, 10% fetal bovine serum (FBS) and 1% penicillin/streptomycin. In glucose labeling experiments, unlabeled glucose was replaced with [$^{13}C_6$]glucose. Similarly, in glutamine labeling experiment, unlabeled glutamine was replaced with [$^{13}C_5$]glutamine. This media was used directly for MIO-M1 or primary Müller cells. For the endothelial cell experiments, this media was diluted 1:1 with Classic Medium containing Serum and CultureBoost™ (Cell systems, Kirkland, WA, USA). For isotopic labeling studies, cells were cultured in 6-well plates to 90% confluence and then media were replaced with media containing labeled substrate(s). Following media change, cells were incubated for 24 h in a normoxic cell culture incubator set at 5% CO₂ and 37 °C, to establish isotopic steady state. Once isotopic steady state was reached, cells were either incubated in a normoxic or hyperoxic (75% O₂ with 5% CO₂) incubator for 8 or 24 h as stated in the Results section.

Human cortical astrocytes were cultured in Astrocyte medium (Sciencell; catalog no 1801). All cells purchased from Sciencell have the following information listed regarding informed consent: "Human tissue used for the isolation of primary cells is derived from donors who have signed informed consent by the donor themselves or an authorized agent acting on the donor's behalf". For labeling experiments, media was replaced with DMEM (without glucose, glutamine and pyruvate) supplemented with 15 mM glucose, 5 mM glutamine, 10 % FBS and 1% penicillin/streptomycin. For glucose labeling experiments, [$^{12}C_6$]glucose was replaced with [$^{13}C_6$]glucose. Similarly, for glutamine labeling experiments, [$^{12}C_5$] glutamine was replaced with [$^{13}C_5$]glutamine. Same media composition was used for primary Müller cell labeling experiment.

**Primary Müller cell isolation and culture**. Enucleated eyes from P11-P12 mice were held in dissecting medium (EBSS + 1% Penicillin/Streptomycin) on ice. Dissected retinas were minced and transferred to a collection tube with minimal volume of dissecting medium, held on ice. Retinal tissue was dissociated using the Worthington Papain Dissociation System (#LK003150). Briefly, a vial of papain was reconstituted in EBSS equilibrated with sterile 95% O₂ : 5% CO₂ and incubated at 37 °C for 10 min until fully dissolved. A vial of DNase I was reconstituted in equilibrated EBSS, and 250 μl was added to the activated papain. This papain/DNase solution was added to the retinal tissue and triturated gently with a P1000 pipette. The tissues were equilibrated in sterile 95% O₂ : 5% CO₂ and incubated at 37 °C with 160 rpm agitation for a total of 35 min, then divided up into 15 min, 15 min, and 5 min incubations, triturating gently after each. The cell suspension was centrifuged at 550 × g for 5 min at room temperature and the pellet resuspended in ovomucoid inhibitor solution, prepared according to the manufacturer's instructions. Cells were again pelleted at 550 × g for 5 min at room temperature and passed through a 30 μm filter after resuspending in 1 ml primary Müller glia cell culture media: DMEM-high glucose, without L-glutamine, with sodium pyruvate + 1% GlutaMax, 1% Penicillin/Streptomycin, and 10% FBS.

Cells obtained from the dissociation were counted, and a trypan blue dye exclusion assay was used to determine a final viability of about 83%. Retinal cells were plated at a density of $1.1 \times 10^6$ cells/ cm² onto plates coated with 10 μg/cm² laminin and maintained and maintained at 37 °C in primary Müller glia cell culture media. The first media change occurred after four days, followed by media changes every two days. Cells were harvested after 8 days in culture, at which point they are primarily comprised of Müller glia.

**Metabolite extraction**. To extract metabolites, the media were aspirated from the 6-well plates with adherent cells and cells were quickly washed with 1ml of room temperature normal saline. To each well, 400 μl of −20 °C cold methanol and 400 μl of cold water was added. Cells were scraped with a cell scraper, while maintaining plates on ice. The resulting cell suspension was added to 400 μl of −20 °C cold chloroform. Tubes were agitated on thermomixer for 20 min at 4 °C and 1400 rpm. Tubes were then centrifuged at 15000 × g for 5 min at 4 °C. 300 μl of upper layer containing polar metabolites was dried under vacuum at −4 °C. In glutamine labeling experiment, 10 μl of 0.05 mg/ml of [$^{13}C_5$]ribitol was added as internal and recovery standard. Dried extracts were derivatized with 25 μl of 40 mg/ml of methoxylamine in pyridine for 30 min on a thermomixer set at 45 °C and 1000 rpm. These samples were further derivatized with 75 μl of MSTFA + 1% TMCS.

**Ammonium extraction/derivatization for GC-MS measurement**. Cells cultured in 6-well plates were either incubated in normoxia or hyperoxia. Ammonium was measured with the method described in Yang et al.[39] with slight modifications[40]. After 48 h of incubation, 400 μl of media sample was taken and added to 400 μl of 10 N NaOH in 1.5 ml tube. To this preparation, 400 μl of formaldehyde (36.5–38% in water) was added and left overnight at room temperature in the fume hood. Formaldehyde reacts with ammonium to produce a stable compound hexamethylenetetramine (HMT). To this HMT preparation, 200 μl of 10% iodine was added to form the extractible HMT-iodine adduct. Tubes were incubated on a thermomixer at 40 °C for 15 min with 1000 rpm agitation speed. Tubes were cooled down to room temperature and the whole mixture was added to 5 ml of chloroform, to extract HMTiodine from the solution. Tubes were centrifuged at 1000 × g at room temperature for 1 min to separate phases. 100 μl of lower phase containing HMTiodine was taken out into a fresh tube and 1 μl of alkane mix (C10-C40 50 mg/L, even chain alkanes) standard was added to the solution as internal standard. Sample 1 μl was then injected into the mass spectrometer and m/z 140 corresponding to mass of HMT was used for quantitation. The HMT area was normalized to that of one of the alkanes with m/z 198, which corresponds to mass of $C_{14}H_{30}$.

**Retinal explant labeling**. Mouse were euthanized with a lethal dose of ketamine/xylazine, eyes enucleated and retinas were dissected in high glucose DMEM. For labeling from [1-$^{13}C$]acetate, retinas were briefly washed with room temperature normal saline and immediately added to 1 ml of culture media in 12-well plates. Media used for the tracer experiment was DMEM containing 5 mM glucose, 1 mM glutamine, 2 mM [1-$^{13}C$]acetate, 10% FBS and 1% Penicillin/Streptomycin. Retinas were cultured for 30 min or 1 h, after which they were washed with room temperature normal saline and then snap frozen in liquid nitrogen. Metabolites from retina were extracted using 500 μl of −20 °C cold 80% methanol, brief sonication and centrifugation at 15,000 × g for 5 min at 4 °C. Supernatant 350 μl was taken into a fresh tube and 10 μl of 0.05 mg/ml of [$^{13}C_5$]ribitol was added to each sample before drying for GCMS measurements. Metabolites were dried, derivatized, measured and data analyzed as described for astrocytes.

For glutamine labeling, retinas were dissected as described above for acetate labeling; dissections were performed in KRB containing 5 mM glucose and 1% penicillin/streptomycin. Retinas were then incubated in KRB media containing 5 mM of [$^{13}C_5$]glutamine, 10% dialyzed FBS and 1% penicillin/streptomycin for 5 min. After 5 min of incubation in labeling media, retinas were removed and washed with 1 ml of normal saline. After this step, 1 ml of KRB media containing 10% FBS, 1% Penicillin/Streptomycin, 5 mM lactate and 5 mM glucose was added to each well of the 12-well plates containing retinal explants. Metabolites were extracted from the retinal explants using methanol/chloroform/water protocol as described earlier. Ten microliters of 0.05 mg/ml [$^{13}C_5$]ribitol was added to 300 μl of supernatant. Samples were dried under vacuum and derivatized as described earlier and measured on GCMS.

**Cell counting**. Media from the cells cultured in 6-well plates in normoxic or hyperoxic incubator were removed. Cells were washed with sterile 2 ml PBS. Cells were trypsinized with 100 μl of 0.05% trypsin containing 0.53 mM EDTA and resuspended in 1 ml of fresh media. Tubes containing cells were centrifuged at 300 × g at room temperature for 5 min. Media were removed and cells were resuspended in 1 ml of HBSS. Cells were again washed and fresh 1 ml of HBSS was added. 50 μl of this was added to 50 μl of 2x CyQuant cell counting reagent and samples were incubated at 37 °C in cell culture incubator for 1 h, following which fluorescence was measured on Victor X2 plate reader (PerkinElmer, Waltham, MA) with excitation at 485 nm and emission detection at 535 nm with exposure time of 0.1 s.

### Table 1 Single ion monitoring (SIM) windows.

| Time window | Metabolite | Ions (m/z) |
|---|---|---|
| 6.0–6.16 | Pyruvate | 173–178 |
| 6.16–9.00 | Lactate | 218–223 |
| 9.0–9.65 | Proline | 243–252 |
| 9.65–10 | Succinate | 246–252 |
| 10–12.06 | Fumarate | 244–250 |
| 12.06–12.40 | Malate | 334–340 |
| 12.40–13.57 | Aspartate | 333–339 |
| 13.57–14.90 | Glutamate | 347–354 |
| 14.90–15.29 | U-$^{13}$C Ribitol | 323, 336–342 |
| 15.29–15.80 | Glutamine | 346–353 |
| 15.80–16.60 | Citrate | 464–474 |
| 16.60–37.50 | Glucose | 553–561 |

Time window, ions measured between these time limits; Metabolites, name of the compounds; Ions (m/z), mass-to-charge ratio measured.

**GC-EI-MS analysis of metabolites**. Derivatives from glucose and glutamine labeling experiment were measured on 7890B GC coupled to EI/CI 5977 mass selective detector Agilent mass spectrometer. Electron impact ionization was used for all the measurements. One μl of sample was injected in splitless mode. Full scan method was used for glucose labeling experiment, with scan widow from 50 to 800 m/z. Front inlet heater was set to 250 °C, septum purge flow 3 ml/min. Injections were made in splitless mode onto HP-5ms inert 30 m × 250 μm × 0.25 μm column connected to MSD. Samples were run in a constant flow mode with flow of helium set to 1.1 ml/min. The oven temperature was set to a starting 60 °C for 1 min and then ramped at 10 °C/min to 325 °C with a final hold time of 10 min. The column was re-equilibrated at 60 °C for 1 min.

Primary Müller cells and primary astrocyte metabolites were assayed separately on a similar column, DB-5 ms GC Column 30 m × 0.25 mm × 0.25 μm with DuraGuard 10 m. Scan method was similar with solvent delay of 6.6 min and Selected ion monitoring (SIM) method used was also similar with solvent delay of 6.6 min. Ions used in SIM method were same with different time windows to account for differences in retention time on the new column.

SIM method was used for metabolite measurement for glutamine labeling assays. SIM windows used are provided below in Table 1.

**Calculations of isotopic enrichments and concentrations of metabolites**. Metabolite Detector and Mass Hunter software were used for deconvolution and annotation of the metabolites[41]. IsoCor was used to correct for natural abundance of isotopes and to derive true mass isotopomer distribution for each metabolite analyzed[42]. Unlabeled extracts from MIO-M1 cells and authentic metabolite standards were used to annotate metabolites in all the samples. NIST 2017 library was further used to confirm the compounds in the extracts from all cell lines and retinal explants.

**Western blot**. MIO-M1 cells were treated similarly as for metabolite labeling experiment, except the media used only contained unlabeled glucose and glutamine. The cells were scraped in RIPA buffer (Sigma-Aldrich, St. Louis, MO, USA) with protease inhibitor cocktail Complete and phosphatase inhibitor cocktail PhosSTOP (both from Roche Diagnostics, Mannheim, Germany). Protein concentration was measured with Pierce BCA protein assay reagent (Thermo-scientific, Bellefonte, PA, USA). Protein 8 or 16 μg were added 20 mM DTT (final concentration), volume was adjusted to 30 μl with 2× Sample Tris-Glycine SDS buffer from Novex Invitrogen (Bellefonte, PA, USA), and were heated at 95 °C for 3 min. The quantity of proteins in all the samples used in a single Western blot were equal. 25 μl of this sample preparation was then loaded onto the Tris-glycine gel 4–20% precast gels (Invitrogen, Bellefonte, PA, USA) and ran at 125 V for 1 h 45 min. Proteins were transferred to PVDF membrane using a wet transfer method and membrane was dried in for 1 h by sandwiching the membrane between two chromatography papers. Membrane was then blocked with Odyssey buffer (LI-COR, Lincoln, Nebraska, USA), for 1 h at room temperature, following which membranes were incubated overnight with primary antibody at a dilution recommended by the vendor. After overnight incubation with primary antibody, membrane was washed with TBST buffer and incubated with secondary antibody for 1 h at room temperature in dark.

Following primary antibodies were used:

(1) PDH E1 component subunit alpha rabbit polyclonal Anti-PDH antibody catalog number ABS2082 from Millipore (Burlington, MA, USA). Dilution used 1:1000.
(2) PhosphoDetect™ Anti-PDH-E1α (pSer232) Rabbit pAb catalog number AP1063 from Millipore (Burlington, MA, USA). Dilution used 1:1000.

(3) COXIV antibody catalog number 4844 from Cell Signaling Technology (Danvers, MA, USA) was used to probe mitochondria in the samples. Dilution used 1:1000.
(4) β-Actin mAbs catalog #3700 and #4970 Cell Signaling Technology (Danvers, MA, USA) as loading control. Dilution used 1:2000.
(5) GS mouse mAb #610517 (BD Bioscience, San Jose, CA)—marker of Müller glia. Dilution used 1:1000.
(6) CRALBP rabbit pAb UW55 originally raised by Jack Saari (University of Washington) and gifted to us by John W. Crabb (Cleveland Clinic). Dilution used 1:1000.

Secondary antibodies used were:

(1) IRDye® 680RD Donkey (polyclonal) anti-mouse IgG (H+L), catalog number 925-68072 from LI-COR (Lincoln, Nebraska, USA) Dilution used 1:1000.
(2) IRDye® 800CW Donkey (polyclonal) anti-Rabbit IgG (H+L), catalog number 925-32213 from LI-COR (Lincoln, Nebraska, USA) Dilution used 1:1000.

**Mitochondria staining**. Mitochondria morphology was revealed using Invitrogen CellLight Reagent BacMam 2.0 Mitochondria-GFP (ThermoFisher), which represents a baculoviral vector construct encoding a signal peptide, i.e., the leader sequence of E1α pyruvate dehydrogenase, fused to emerald GFP. MIO-M1 or REC cells were grown at 70% confluence in four-well Nunc Lab-Tek II Chamber Slides (ThermoFisher) were transfected overnight with Mitochondria-GFP at $3 \times 10^5$ or $10 \times 10^5$ particles/well, respectively. Next day cells continued to culture in the regular humidified $CO_2$ incubator or were placed in the humidified hyperoxic cell culture chamber with 75% oxygen and 5% $CO_2$ conditions controlled by ProOx Model C21 (Biospherix, Parish, NY) and connected to the sources of $CO_2$ and $O_2$. After overnight incubation cells were quickly rinsed with warm PBS, fixed in warm freshly prepared 4% paraformaldehyde in PBS, and mounted in VectaShield with DAPI (Vector Labs, Burlingame, CA). Fluorescent images were taken using a Zeiss AxioImager.Z1 fluorescent microscope equipped with 63× objective, AxioCam 503 mono camera and ApoTome.2 adapter. Quantitative analyses of mitochondrial networks were performed using two custom macros for NIH ImageJ v1.40 g image analysis software Mitophagy and MiNA[43,44].

**Reporting summary**. Further information on research design is available in the Nature Research Reporting Summary linked to this article.

## Data availability
All relevant data related to this manuscript are provided in the source data file. Metabolomics data are available online via metabolights study ID MTBLS1228. The source data underlying all figures are provided.

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

## Acknowledgements

The immortalized human Müller cell line Moorfields/Institute of Ophthalmology-Müller 1 (MIO-M1) was obtained from the UCL Institute of Ophthalmology, London, UK through a generous gift of Dr Astrid Limb. Primary human cortical astrocytes were a generous gift from Dr Jessica Williams. We are grateful for the contributions of Demiana Hanna and Andrew Benos. Grant Support: National Eye Institute (R01 EY024972 to JES; P30 EY025585 to Ophthalmic Research, T32 5T32EY024236-04 to KA); The Hartwell Foundation Biomedical Research Fellowship (HWF06092015 to JES); Research to Prevent Blindness Physician Scientist (RPB1801 to JES).

## Author contributions

C.S., J.E.S. and H.B conceived the experiments, interpreted the data and wrote the paper. C.S., V.T., L.M., Y.B., and G.H. performed the experiments. K.A. and A.Y. provided primary cultured Müller cells. G.H. and H.B. edited the paper.

## Competing interests

The authors declare no competing interests.
