## [Peer Review File · Nature Communications]

Reviewers' comments:

Reviewer #1 (Remarks to the Author):

This paper reports interesting effects of hyperoxia on cultured retinal endothelial cells and cultured Müller cells. The steady state measurements of incorporation of ¹³C label from glucose and glutamine are very thoroughly done and clearly reported and the effects of hyperoxia are interesting, substantial and different between endothelial cells and cultured Müller cells.

Although the paper is technically sound, my main concern with the significance of the findings is that the studies are done almost entirely with cultured cell lines. It is very likely, based on reported studies, that metabolic features of these cell lines may not accurately reflect metabolic features of endothelial cells and Müller cells in their native environment in a retina. In my opinion, this report would be stronger if it included data that shows how accurately the metabolism of these cultured cells can represent the metabolism of the cells that they are being used to represent when those cells are in an intact retina. For example, how does expression of known Müller cell markers like CRALBP or glutamine synthetase compare to the levels of expression of those Müller cell specific proteins in a retina (for example, compare the amount of those proteins (by quantitative immunoblotting) per ug protein in the cultured cell vs in the retina). I noted that the paper in its current form does include (near the end of the results section) some studies using intact retinas so I would support the idea of including more experiments like that to bolster the significance of the findings. It would be helpful to find more ways to take the interesting effects of hyperoxia that the investigators discovered in cultured cells and compare them in detail to the effects on cell-type specific metabolic activities of those cells in their native environment in intact retinas.

Minor comment:

1. I think that the statement p. 11: "PKM1 is present in photoreceptors" is inaccurate. Several relatively recent reports from multiple labs indicate that PKM2 is the isoform expressed in photoreceptors. At least one study has reported data suggesting that Müller cells in intact retinas may not express any isoform of PK. If that is correct, it also demonstrates a substantial difference between Müller cells in the retina vs the cultured cell lines.

James B. Hurley
University of Washington

Reviewer #2 (Remarks to the Author):

In their manuscript titled "Hyperoxia induces glutamine-fueled anaplerosis and reverses glutamine/glutamate cycling in retinal Müller cells," Singh et al. compare an immortalized cell line made from human retina to vascular endothelial cells in normoxic and hyperoxic conditions. In particular, they compare metabolic processes related to glucose and glutamine metabolism. They report dramatic differences in metabolism between the cell lines under hyperoxic conditions. For example, they show a dramatic decrease in entry of glycolytic carbon into the TCA cycle in the MIO-M1 cell line relative to the vascular endothelial cells. The major limitations of the study are related to the cells used and the validation of the findings. Specifically:

- 1) The authors claim the MIO-M1 cells are Müller glia but this has not been rigorously demonstrated in the literature. Indeed, the cells have been reported to have stem/progenitor cell properties.
- 2) Findings should have been validated in Müller glia in vivo using mice or other species.
- 3) It is not ideal to rely on only one cell line for each cell type being studied as this could lead to artifacts.
- 4) The mitochondrial analyses are preliminary at best and there are many assays available to monitor mitochondrial mass and function in cells.

Dear Dr. Parrish,

Thank you for the constructive criticisms regarding our manuscript, "Hyperoxia induces glutamine-fueled anaplerosis in retinal Müller cells," by Singh et al. We have addressed the criticisms from both the reviewers by recapitulating our findings using stable isotope resolved metabolomics in primary Müller cells, in primary cortical brain glia, and in additional studies of intact retinal explants. We have also quantified expression of key marker proteins to relate the how cells *in vitro* are like cells *in vivo* and quantified mitochondrial density. A point by point response is provided below. If requested, we will email you an invitation to view our raw and processed data via our labarchives electronic lab notebook (ELN) so that the same can be shared with reviewers if needed. ELN is new to our research department and its public sharing feature is still under construction. All metabolomics data has also been deposited to Metabolights (study ID MTBLS1228) and will be available once curated and reviewed by Metabolights curators.

Reviewer #1 (Remarks to the Author):

NB: references listed below are presented at the end of this text

This paper reports interesting effects of hyperoxia on cultured retinal endothelial cells and cultured Müller cells. The steady state measurements of incorporation of ¹³C label from glucose and glutamine are very thoroughly done and clearly reported and the effects of hyperoxia are interesting, substantial and different between endothelial cells and cultured Müller cells.

Although the paper is technically sound, my main concern with the significance of the findings is that the studies are done almost entirely with cultured cell lines. It is very likely, based on reported studies, that metabolic features of these cell lines may not accurately reflect metabolic features of endothelial cells and Müller cells in their native environment in a retina. In my opinion, this report would be stronger if it included data that shows how accurately the metabolism of these cultured cells can represent the metabolism of the cells that they are being used to represent when those cells are in an intact retina. For example, how does expression of known Müller cell markers like CRALBP or glutamine synthetase compare to the levels of expression of those Müller cell specific proteins in a retina (for example, compare the amount of those proteins (by quantitative immunoblotting) per ug protein in the cultured cell vs in the retina). I noted that the paper in its current form does include (near the end of the results section) some studies using intact retinas so I would support the idea of including more experiments like that to bolster the significance of the findings. It would be helpful to find more ways to take the interesting effects of hyperoxia that the investigators discovered in cultured cells and compare them in detail to the

effects on cell-type specific metabolic activities of those cells in their native environment in intact retinas.

Reviewer 1:

1. In my opinion, this report would be stronger if it included data that shows how accurately the metabolism of these cultured cells can represent the metabolism of the cells that they are being used to represent when those cells are in an intact retina. For example, how does expression of known Müller cell markers like CRALBP or glutamine synthetase compare to the levels of expression cell markers to those Müller cell specific proteins in a retina (for example, compare the amount of those proteins (by quantitative immunoblotting) per ug protein in the cultured cell vs in the retina).

We have quantified expression of cellular retinaldehyde binding protein (CRALBP) and glutamine synthetase (GS) in cultured human Müller cells (MIO-M1) and normalized signal from western blot to either β -actin expression or total μ g protein per lane (Figure S3A-B). Quantification demonstrates that, as the reviewers' suspected, whole retina and primary mouse Müller cells express more CRALBP and GS than immortalized Müller cells. This reinforces the advice from both reviewers that stable isotope based metabolomics be repeated using other cell/tissue types such as primary Müller cells, intact retina, and additional glial cell types, such as primary cortical astrocytes. Additional studies added to the manuscript using primary cultured Müller cells and whole retina demonstrate a robust expression of these Müller cell specific markers confirm that primary cells and retina are good models to definitively demonstrate hyperoxia-induced glutamine-fueled anaplerosis, first found in MIO-M1 cells. These findings also align with reports in the literature regarding the "faithfulness" of MIO-M1(1). Our primary Müller cells have the highest GS/CRALBP ratio, and of the 3 different cells in culture behave closest to glia in whole retina. One might imagine that pool sizes of glutamine would differ given different expression patterns of GS, yet we find that percentage change in fractional enrichment is relatively the same for cells in the same conditions, further definitively demonstrating that hyperoxia-induced glutamine-fueled anaplerosis is a real biological phenomenon, as we demonstrate that it occurs in retinal explants, MIO-M1 cells, primary Müller cells, and primary brain cortical astrocytes. We also find it interesting that primary Müller cells predominantly use oxidative decarboxylation once α -ketoglutarate (α KG) is formed from glutamine. A second difference we note is that although primary astrocytes also oxidize glutamine in hyperoxia, metabolites downstream of α -KG do not accumulate as they, unlike Müller cells, have a mitochondrial AGC1 transporter that allows exchange of aspartate and glutamate between cytosol and mitochondria. We have added Supplemental Figure 3A,B, and refer to this figure on page 7, line 8,

"To determine whether oxygen induced glutamine-fueled anaplerosis described above in immortalized human Müller cells also occurs in primary Muller cells, we isolated primary muller cells from P11 mice. Lysates of MIO-M1 Müller cells, primary Müller cells, retinal explants, and primary human astrocyte cultures compared by western blotting for protein levels of glutamine synthase and cellular retinaldehyde

binding protein (CRALBP) to ensure that cultured cells were differentiated glia (Fig. S3A,B). Primary Müller cells expressed similar ratios of CRALBP/GS as was found in glia from retinal explants.”

2. I noted that the paper in its current form does include (near the end of the results section) some studies using intact retinas so I would support the idea of including more experiments like that to bolster the significance of the findings. It would be helpful to find more ways to take the interesting effects of hyperoxia that the investigators discovered in cultured cells and compare them in detail to the effects on cell-type specific metabolic activities of those cells in their native environment in intact retinas.

In order to demonstrate the effect of hyperoxia on Müller cell metabolism in whole retinal explants, we continued our experiments on retinal explants using metabolomics to follow label from [1-¹³C] acetate. This approach confirmed that there is a decrease in acetate entry into the TCA cycle. We have added the following text, page 11, paragraph 4, and provided additional Figures 3L and 3M:

“Waniewski and Martin, and others, have reported higher acetate utilization by glia/astrocytes than by neurons and that acetate is metabolized in retina in the same way as in the brain (2-7). We used [1-¹³C] acetate to study the effect of hyperoxia on Müller cell metabolism in cultured retinal explants. Retina from p10 old mice were dissected and subsequently cultured in DMEM containing glucose, [1-¹³C] acetate and glutamine (Fig. 3L). Retinas were exposed to normoxia or hyperoxia for 30 min or 1h, following which metabolites were extracted and measured. As in isolated cell culture experiments, as expected, fractional enrichment of M1 Citrate in retinal explants was decreased and M2 Aspartate was increased in response to hyperoxia at 30 minutes and 1 hour (Fig. 3M).”

3. I think that the statement p. 11: “PKM1 is present in photoreceptors” is inaccurate. Several relatively recent reports from multiple labs indicate that PKM2 is the isoform expressed in photoreceptors. At least one study has reported data suggesting that Müller cells in intact retinas may not express any isoform of PK. If that is correct, it also demonstrates a substantial difference between Müller cells in the retina vs the cultured cell lines.

We have changed this statement, (now page 13, line 10, to read “PKM1/PKM2 is present in photoreceptors, whereas weak expression of the PKM2 is expressed in other layers of retina.”). Regarding whether or not Müller cells express PKM, I think the additional question why glycolytic carbon does not enter TCA in hyperoxia is presently without an answer and on its own may well be a separate body of work. If the reviewers would prefer removing all supplemental figures regarding the mitochondria, I will comply. Thus far, we only found the following about mitochondria in hyperoxia:

1) PKM1 is expressed in normoxia and hyperoxia and 2) there is equal phosphorylation states of PDH, comparing hyperoxia to normoxia, 3) that COXIV protein levels are not affected by hyperoxia implying no difference in mitochondrial quantity, 4) that there is no

oxygen-induced decrease in acetylCoA or CoA concentrations and that 5) citrate synthase activity is not decreased in hyperoxia. We are working to discover the mechanism through which hyperoxia affects the flow of glycolytic carbon into the TCA. We do not have the answer yet and I agree that this isn't the focus of this manuscript and might be better developed more completely as an independent study.

We are very grateful for the comments of Reviewer 1.

Reviewer 2.

In their manuscript titled “Hyperoxia induces glutamine-fueled anaplerosis and reverses glutamine/glutamate cycling in retinal Müller cells,” Singh et al. compare an immortalized cell line made from human retina to vascular endothelial cells in normoxic and hyperoxic conditions. In particular, they compare metabolic processes related to glucose and glutamine metabolism. They report dramatic differences in metabolism between the cell lines under hyperoxic conditions. For example, they show a dramatic decrease in entry of glycolytic carbon into the TCA cycle in the MIO-M1 cell line relative to the vascular endothelial cells. The major limitations of the study are related to the cells used and the validation of the findings. Specifically:

1) The authors claim that MIO-M1 cells are Müller glia but this has not been rigorously demonstrated in the literature. Indeed, the cells have been reported to have stem/progenitor cell properties.

Müller cells have been reported to have stem and progenitor cell properties, but this has not been reported for unstimulated MIO-M1 cells outside of the fact they are spontaneously immortalized cells. Nevertheless immortalization is certainly a characteristic of stem cells, and spontaneously immortalized human cell lines have been cultured that express neuronal cell markers and form neurospheres when exposed to extracellular matrix and fibroblast growth factor-2 or retinoic acid (8). In the mouse, stem-like properties of Müller cells have been described when they are stimulated by a specific combination of overexpression of the transcription factor ASCL1 and an HDAC inhibitor *in vitro* (9). But I suspect that placing Müller cells in hyperoxia will not cause them to become pluripotent and morphologically these cells retain a stellate shape characteristic of Müller cells in culture. But as a means of demonstrating the validity of using these cells, we have quantified GS and CRALBP expression in MIO-M1, primary Müller cells, retinal extracts, and primary cortical astrocytes, and found that primary Müller cells and retinal extracts express Müller cell markers to a greater extent than MIO-M1 and brain cortical astrocytes (Supplemental Figures 3A-C). Repeat measurements using stable isotope based metabolomics confirms the observation that in hyperoxia, there is increased glutamine-fueled anaplerosis for primary Müller cells (Fig. 1I), and astrocytes as well (Fig. 1J). Moreover, we used [1-¹³C] acetate to demonstrate that retinal explants in hyperoxia recapitulate the same findings as were observed for primary Müller cells, MIO-M1 cells, and astrocytes (Fig. 3L,M).

2) Findings should have been validated in Müller glia in vivo using mice or other species.

We hope that our results using [1-¹³C] acetate in retinal explants might satisfy this request (Fig. 3L,M). As we describe above, acetate is uniquely metabolized by astrocytes in the central nervous system and by Müller cells in retina. Our isotopic based measurements demonstrate rapid incorporation of label in the tissue before hyperoxia, but once in hyperoxia at 30 minutes and one hour, M1 citrate decreases whereas aspartate (metabolite downstream of α -KG) increases. This demonstrates that before these retinas were placed in hyperoxia, a few rounds of TCA cycle already took place leading to the formation of M2 OAA/Aspartate. When these cells are placed in hyperoxia, M2 OAA/Aspartate increases as these cannot be further converted to form M2/M3 citrate. Similar findings have been reported by Ewald et al. where the authors knocked out enzymatic genes in yeast and observed accumulation of substrates in close proximity to deletion of enzymatic genes (10). As an additional mechanism of answering this criticism, I have changed the title of the paper by removing "...and reverses glutamine/glutamate cycling..." so that it reads, "Hyperoxia induces glutamine-fueled anaplerosis in retinal Müller cells."

3) It is not ideal to rely on only one cell line for each cell type being studied as this could lead to artifacts.

We have included additional experiments using primary mouse Müller cells (Fig. 1I), primary human brain cortical astrocytes (Fig. 1J), and whole mouse retinal explants (Fig. 3J,K,L,M), along with previous studies of retinal endothelial cells (Fig. 2). All isotopic measurements definitively demonstrate hyperoxia-induced decreased entry of glucose into the TCA and increased glutaminolysis in retinal explants, MIO-M1 cells, primary Müller cells, and primary astrocytes. As we have described above, although in all the cell lines entry of glycolytic carbon into the TCA cycle is decreased and glutaminolysis is increased, astrocytes have a diluted pool of aspartate and fumarate owing to dilution of labeled aspartate exchanged between cytosol and mitochondria due to the presence of mitochondrial AGC1 transporter protein. This transporter is absent in Müller cells and therefore aspartate cannot leave the mitochondria. Primary Müller cells use oxidative decarboxylation to a greater extent than MIO-M1 cells.

4) The mitochondrial analyses are preliminary at best and there are many assays available to monitor mitochondrial mass and function in cells.

We have re-evaluated our former immunohistochemistry slides and provided a mitochondria density comparison of normoxia to hyperoxia in the revised version of the manuscript. We have quantified mitochondrial density and provide the results in Supplemental Figure 6, and added the following text, page 13, line 7," Quantification of mitochondrial morphology in both RECs and Müller cells is provided in Supplemental Figure 6E-H. It demonstrates equal density of mitochondria but definite change in morphology due to hyperoxia in Müller cells and RECs, which have clumped

mitochondria in hyperoxia that represent smaller fragmented and reduced branching of mitochondrial networks. ”

I appreciate very much the comments of Reviewer 2.

Both reviewers had similar, valuable suggestions about this investigation, and I am hopeful that providing data on Müller cell marker expression patterns, stable isotope resolved metabolomics data on two additional primary cell lines, and additional retinal explant experiments will be convincing that hyperoxia induces glutamine-fueled anaplerosis in cells that ordinarily synthesize, not catabolize glutamine.

Sincerely,

Jonathan Sears

1. Limb GA, Salt TE, Munro PM, Moss SE, Khaw PT. In vitro characterization of a spontaneously immortalized human Muller cell line (MIO-M1). *Investigative ophthalmology & visual science*. 2002;43(3):864-9. Epub 2002/02/28. PubMed PMID: 11867609.
2. Waniewski RA, Martin DL. Preferential utilization of acetate by astrocytes is attributable to transport. *J Neurosci*. 1998;18(14):5225-33. Epub 1998/07/03. PubMed PMID: 9651205; PMCID: PMC6793490.
3. Patel AB, de Graaf RA, Rothman DL, Behar KL, Mason GF. Evaluation of cerebral acetate transport and metabolic rates in the rat brain in vivo using ¹H-[¹³C]-NMR. *J Cereb Blood Flow Metab*. 2010;30(6):1200-13. Epub 2010/02/04. doi: 10.1038/jcbfm.2010.2. PubMed PMID: 20125180; PMCID: PMC2879471.
4. Deelchand DK, Shestov AA, Koski DM, Ugurbil K, Henry PG. Acetate transport and utilization in the rat brain. *J Neurochem*. 2009;109 Suppl 1:46-54. Epub 2009/05/07. doi: 10.1111/j.1471-4159.2009.05895.x. PubMed PMID: 19393008; PMCID: PMC2722917.
5. Hosoi R, Matsuyama Y, Hirose S, Koyama Y, Matsuda T, Gee A, Inoue O. Characterization of (¹⁴C)-acetate uptake in cultured rat astrocytes. *Brain Res*. 2009;1253:69-73. Epub 2008/12/17. doi: 10.1016/j.brainres.2008.11.068. PubMed PMID: 19073161.
6. Virgili M, Paulsen R, Villani L, Contestabile A, Fonnum F. Temporary impairment of Muller cell metabolism in the rat retina by intravitreal injection of fluorocitrate. *Exp Eye Res*. 1991;53(1):115-22. Epub 1991/07/01. doi: 10.1016/0014-4835(91)90153-6. PubMed PMID: 1879495.
7. Starr MS. A comparative study of the utilization of glucose, acetate, glutamine and GABA as precursors of amino acids by retinal of the rat, frog, rabbit and pigeon. *Biochem Pharmacol*. 1975;24(11-12):1193-7. Epub 1975/06/15. doi: 10.1016/0006-2952(75)90061-1. PubMed PMID: 1079726.

8. Lawrence JM, Singhal S, Bhatia B, Keegan DJ, Reh TA, Luthert PJ, Khaw PT, Limb GA. MIO-M1 cells and similar muller glial cell lines derived from adult human retina exhibit neural stem cell characteristics. *Stem Cells*. 2007;25(8):2033-43. Epub 2007/05/26. doi: 10.1634/stemcells.2006-0724. PubMed PMID: 17525239.
9. Jorstad NL, Wilken MS, Grimes WN, Wohl SG, VandenBosch LS, Yoshimatsu T, Wong RO, Rieke F, Reh TA. Stimulation of functional neuronal regeneration from Muller glia in adult mice. *Nature*. 2017;548(7665):103-7. Epub 2017/07/27. doi: 10.1038/nature23283. PubMed PMID: 28746305; PMCID: PMC5991837.
10. Ewald JC, Matt T, Zamboni N. The integrated response of primary metabolites to gene deletions and the environment. *Mol Biosyst*. 2013;9(3):440-6. Epub 2013/01/24. doi: 10.1039/c2mb25423a. PubMed PMID: 23340584.

Reviewers' comments:

Reviewer #1 (Remarks to the Author):

The authors satisfactorily addressed my previous concerns.
The following minor points should be checked.

1. All cases of "glutamine synthase" should be "glutamine synthetase".
2. Please check lines 277-278 – it seems like words might be missing.

James B. Hurley

Reviewer #2 (Remarks to the Author):

While the metabolomics work is solid, my major concern is the source of the cells. Throughout the manuscript, the authors refer to the immortalized cells as Muller glia. There is very little evidence that those cells are Müller glia and there has been no determination of the impact of culturing and immortalization on the metabolism in those cells. They should not use the term Muller glia throughout the manuscript as it is confusing. They should refer to the MIO-M1 cell line when it is used. Expression of a few markers is not sufficient to validate the identity. There are now very nice single cell RNA-seq datasets available from published literature that could have been used to validate the cell line. Also, morphology is an important consideration as Muller glia have a unique structure. Finally, quiescent Muller glia are very different from those undergoing reactive gliosis and if their cell line expresses GFAP, then it is reactive so the relevance to normal Muller glia physiology is questionable. The authors did some studies on dissociated retina from P11-P12 but those are not mature. They should have used adult retina and that was a mixed culture of all retinal cell types not just Muller glia. Thus, the retinal experiments are not particularly useful.

Reviewers' comments:

Reviewer #1 (Remarks to the Author):

**The authors satisfactorily addressed my previous concerns.
The following minor points should be checked.**

- 1. All cases of “glutamine synthase” should be “glutamine synthetase”.**
- 2. Please check lines 277-278 – it seems like words might be missing.**

James B. Hurley

We want to thank Reviewer 1 for constructive criticisms, and that our additional experiments have convinced reviewer 1 of the veracity of our findings.

Additionally, we have replaced “glutamine synthase” with “glutamine synthetase”.

We have corrected the typos (lines 277-278) in the updated version.

Reviewer #2 (Remarks to the Author):

While the metabolomics work is solid, my major concern is the source of the cells. Throughout the manuscript, the authors refer to the immortalized cells as Muller glia. There is very little evidence that those cells are Müller glia and there has been no determination of the impact of culturing and immortalization on the metabolism in those cells. They should not use the term Muller glia throughout the manuscript as it is confusing. They should refer to the MIO-M1 cell line when it is used. Expression of a few markers is not sufficient to validate the identity. There are now very nice single cell RNA-seq datasets available from published literature that could have been used to validate the cell line. Also, morphology is an important consideration as Muller glia have a unique structure. Finally, quiescent Muller glia are very different from those undergoing reactive gliosis and if their cell line expresses GFAP, then it is reactive so the relevance to normal Muller glia physiology is questionable. The authors did some studies on dissociated retina from P11-P12 but those are not mature. They should have used adult retina and that was a mixed culture of all retinal cell types not just Muller glia. Thus, the retinal experiments are not particularly useful.

- 1. While the metabolomics work is solid, my major concern is the source of the cells.**

We understand that Reviewer 2 does not trust MIO-M1 cells. This cell line was originally developed by Astrid Limb who indeed published the “faithfulness” of these cells to Müller cells *in vivo*, and yet in collaboration with a retinal regeneration expert, Tom Reh, observed that these very same cells can be induced to become pluripotent. This is in fact the basis to publications that seek to regenerate mammalian retina from existing *in situ* Müller cells. However, this change to pluripotency requires specific exogenous growth factors that include retinoic acid and overexpression of ASCL1, a transcription factor, as well as an HDAC inhibitor. Despite the fact that none of these factors are found in our experiments, we have made sure to perform additional experiments on primary Müller cells, primary human brain cortical astrocytes, and retinal explants as requested by Reviewer 2 in addition to the experiments provided in the original version. Using primary cells in culture, we found similar metabolic changes associated

to hyperoxia in the immortalized and primary cultures. Primary Müller cells were isolated from P11 retina because this is the age at which hyperoxia is used to replicate human retinopathy of prematurity in the mouse model of oxygen induced retinopathy. We have added the following experiments to the revised version, (page 7, line 3), and as a convenience, I have appended relevant Figures to make review easier:

Beginning on page 7, line 3:

“Glutamine labeling of primary Müller cells also exhibit increased glutamine utilization and drop in glycolytic carbon entering TCAC

To determine whether oxygen induced glutamine-fueled anaplerosis described above in MIO-M1 cells also occurs in primary Müller cells, we isolated primary Müller cells from P11 mice. Lysates of MIO-M1 cells, primary Müller cells, retinal explants, and primary human astrocyte cultures compared by western blotting the levels of glutamine synthetase and cellular retinaldehyde binding protein (CRALBP) to ensure that cultured Müller cells and astrocytes were differentiated glia (Fig. S3A, B). Primary Müller cells expressed similar ratios of CRALBP/GS as was found in glia from retinal explants. Primary Müller cells were cultured in 12-well plates and then incubated in media containing [¹³C₅]glutamine for 24h to establish isotopic steady state. After 24h, cells were incubated either in normoxic or hyperoxic incubator for another 24h. Intracellular metabolites were extracted and measured on GCMS. Like cultured human Müller cells, we saw similar reduction in the proportion of citrate in hyperoxia, implying reduction in flux entering from glycolysis to TCAC (Fig. 1I). In addition, we also found increased M5 glutamate, M4 fumarate and M4 aspartate consistent with oxygen induced increased glutaminolytic flux (Fig. 1I). These findings corroborate well our findings in MIO-M1 cells. Primary Müller cells differ from MIO-M1 cells in that they have a higher ratio of M4/M5 citrate in hyperoxia (Fig. 1I).”

Note that citrate is decreased, yet with carbon labeled glutamine, downstream metabolites of the TCAC such as Fumarate and Aspartate are increased, indicating decreased citrate production and glutamine fueled anaplerosis or the addition of glutaminolytic carbon to the TCAC.

“Primary astrocytes also oxidize glutamine in hyperoxia but have different metabolic plasticity as compared to Müller cells

To further determine whether glutamine-fueled anaplerosis might occur in other types of glia within the central nervous system, we used [$^{13}\text{C}_5$]glutamine to study the effect of hyperoxia on glutaminolytic flux in primary cortical astrocytes. Cells were again cultured in 6-well plates and then incubated in [$^{13}\text{C}_5$]glutamine for 24h to establish isotopic steady state, after which cells were incubated into normoxic or hyperoxic incubators for another 24h. As with cultured human MIO-M1 and primary mouse Müller cells, M5 glutamate was statistically significantly higher in hyperoxic condition, implying higher rate of glutaminolysis in hyperoxia (Fig. 1J). However, primary astrocytes exhibit an interesting difference in the accumulation of metabolites downstream of αKG . In contrast to all Müller cell lines, M4 aspartate and M4 fumarate were lower in hyperoxic condition (Fig. 1J). This observation can be explained by the fact that astrocytes but not Müller cells express the AGC1 transporter protein which allows aspartate and glutamate exchange between mitochondria and cytosol⁴. The difference in M4 aspartate and M4 fumarate enrichments in response to hyperoxia also might be due to decrease in partial isotopic dilution by cytosolic aspartate derived from proteolysis.”

We have additionally used retinal explants to prove our findings from our cell culture experiments, specifically to take into account complex metabolic exchanges between different cells types in the retina. We have added the following, (page 11, paragraph 3), and append the relevant Figure here to make review easier:

Beginning on page 11, paragraph 3:

“Waniewski and Martin, and others, have reported higher acetate utilization by glia/astrocytes than by neurons and that acetate is metabolized in retina in the same way as in the brain¹⁵⁻²⁰. We used [$1\text{-}^{13}\text{C}$] acetate to study the effect of hyperoxia on Müller cell metabolism in cultured retinal explants. Retina from P10 mice were dissected and subsequently cultured in DMEM containing glucose, [$1\text{-}^{13}\text{C}$] acetate and glutamine (Fig. 3L). Retinas were exposed to normoxia or hyperoxia for 30 min or 1h, following which metabolites were extracted and measured. As in isolated cell culture experiments, fractional enrichment of M1 Citrate in retinal explants was decreased and M2 Aspartate was increased in response to hyperoxia at 30 minutes and 1 hour (Fig. 3M).”

2. Throughout the manuscript, the authors refer to the immortalized cells as Muller glia.

Although, we found that the primary Müller cells and MIO-M1 have similar metabolic response to hyperoxia, we have called out MIO-M1 in the article whenever results pertaining to MIO-M1 were presented.

3. Expression of a few markers is not sufficient to validate the identity.

We chose CRALBP and glutamine synthetase, two glia cell markers, at the recommendations of the first review. We did find differences in the expression of these two key Müller cell markers that actually validate our findings because it shows that primary Muller cells and retinal explants express the highest levels of these markers. The supplemental Figures that describe these changes with Figure legends are provided below to make rereview easier:

4. The authors did some studies on dissociated retina from P11-P12 but those are not mature. They should have used adult retina and that was a mixed culture of all retinal cell types not just Muller glia. Thus, the retinal experiments are not particularly useful.

Retinopathy of Prematurity accounts for 200,000 cases of new infant blindness world-wide, and is caused by oxygen supplementation necessary to resuscitating severely premature infants that unfortunately causes retinovascular growth attenuation and vasoobliteration that is the hallmark of phase 1, which subsequently leads to profound ischemia and abnormal angiogenesis in phase 2. Therefore, we chose specifically the hyperoxic phase 1 in the experimental correlate of ROP, the murine oxygen induced retinopathy model (OIR) to test both primary Müller cells and retinal explants. Hyperoxic phase 1 in the mouse model of OIR is from P7 to P12, and to be consistent with the model we have only used cells or retinal explants from mice within the phase 1 of the model. Adult mice don't develop ROP naturally, and therefore using adult retinal explants or isolating primary Müller cells from adult mice will be at odds with the right model to understand the metabolic basis of retinopathy of prematurity.

Summary:

Our paper has one referee who says accept and a second who is obviously in a different camp! But Reviewer 2 does not address the additional experiments we performed at his/her own suggestion to demonstrate that hyperoxia decreases entry of acetate into the tricarboxylic acid cycle (TCAC) and instead adds carbon to the TCAC from glutaminolysis. This is critical for two reasons: 1) premature infants are exposed to excess oxygen and develop the most common form of infant blindness called retinopathy of prematurity (ROP), which blinds 200,000 infants world-wide, and 2) the reversal of normal glutamine metabolism from synthesis to consumption in glia (here we show this metabolic change in immortalized human Müller glia, primary Müller cells, primary human brain cortical astrocytes, and retinal explants) is absolutely fascinating because it demonstrates that hyperoxia is not the reverse (metabolically) of hypoxia. Additionally, it is to note that the Müller cells are the only cell types present in the retina to recycle glutamate to glutamine or to de novo synthesize glutamine for all other cell types in the retina. The very interesting result that hyperoxia changes flux of carbon away from glucose to carbon from glutamine while blocking citrate formation is a very basic, completely novel finding. Unlike hypoxia, which occurs naturally on earth and is well studied, hyperoxia is really only found in oxygen supplementation for severely premature infants and hyperbaric oxygen therapy. Our previous publications in PNAS and JCI Insight have demonstrated the transcriptional and metabolic consequences of hyperoxia and hypoxiamimesis (HIF stabilization). Two metabolic pathways upregulated by HIF stabilization (hypoxiamimesis) and down-regulated by hyperoxia are the urea cycle and 1 carbon metabolism. The former pathway is so interesting in light of the fact that glutamine-fueled anaplerosis or the addition of carbon to the TCAC from glutamine also releases ammonium (NH₄⁺); roughly 2 moles of ammonium for every mole of glutamine that is converted to α KG. Hence the connection to urea formation found in a separate, independent study is intriguing.

Reviewer 2 also missed that we are studying how oxygen changes the developmental course of the retina, as hyperoxia is well-known to cause retinovascular growth attenuation and vasoobliteration which secondarily leads to abnormal angiogenesis. This is the point of not using adult mouse tissues, but rather cells and tissues from the hyperoxic phase of the mouse model of oxygen induced retinopathy (OIR), the experimental correlate of human ROP.

We specifically used acetate as a substrate in our retinal explant experiments. Acetate is only taken up by Müller cells in retina and not by any other cell types. We did not use labeled glucose in our retinal explant experiment to avoid background noise from other cell types.

Once again, our lab is extremely grateful that you are giving us your time and consideration. Reviewer 2 has made important criticisms of our work; we have taken these criticisms earnestly to heart through additional experiments that satisfy Reviewer 2's original and most recent comments. All of the changes that we observe in retinal explants, MIO-M1, primary Müller cells, and primary human cortical brain astrocytes are metabolic phenotype in response to hyperoxia. Conditions for these experiments varied only by the presence or absence of hyperoxia, and led to the same outcome: hyperoxia inhibits entry of glycolytic carbon into the TCAC and instead induces glutamine-fueled anaplerosis.

Thank you, again, for your time and consideration.

Sincerely,

Jonathan Sears

REVIEWERS' COMMENTS:

Reviewer #3 (Remarks to the Author):

I was asked to focus my review on the degree to which the authors have responded to the comments of reviewer 2.

1. The reviewer raises the very important point that the immortalised cells may differ in their behaviour and/or metabolism to primary Müller glial cells. In my opinion, the authors have been robust in their attempts to address this criticism. Most importantly, they have used primary (mouse) Müller glial cells to confirm their findings. They also have performed additional experiments using primary human brain cortical astrocytes and retinal explants.

2. The reviewer requested that throughout the manuscript the authors should refer to the M10-M1 cell line when it is used (rather than referring to them as Müller glia). This has been done.

3. The reviewer quite correctly states that expression of a few markers is not sufficient to validate the identity of the M10-M1 cells. Validation by the authors has been done using only a couple of markers. However, as noted above, the results have been confirmed using primary cells, demonstrating good functional equivalence between the immortalised cells and primary Müller glia cells. The authors also mention in their rebuttal that the cells have been characterised previously by others. However, I did not see any mention of this in the manuscript (apologies if I missed it). I therefore have a couple of suggestions for very minor text changes that would help to further address points 1-3 raised by the reviewer:

(i) In the abstract it would be helpful to clarify that both primary cells and a cell line have been used. For example, the 3rd sentence of the abstract could be modified along these lines: Using a stable isotope labelling technique in human RECs, mouse Müller glial cells and a human Müller glia cell line (M10-M1 cells),.....

(ii) At the start of the results section (for example, after 1st sentence of paragraph 1 of the results), it would be useful to include a short statement explaining what M10-M1 cells are (an immortalised Müller glia cell line) and citing previous work demonstrating the "faithfulness" of these cells to Müller cells (for example the work by Astrid Limb mentioned in the rebuttal). The work done by the authors to characterised the cells themselves also could potentially be mentioned here.

(iii) At the start of the Discussion it would be helpful to again clarify that data are from both primary Müller glia and M10-M1 cells (e.g. Our data obtained using both mouse Müller glial cells and M10-M1 cells (an immortalised human Müller glia-like cell line), demonstrate that....

4. The reviewer questioned the use of dissociated retinal cells from P11-P12 mice, and states that adult retina should have been used instead. I agree with the authors that P11-P12 retina is the appropriate age to have been used in their experiments given the question being addressed. Retinopathy of prematurity is a problem that affects developing, not mature retinas. Using P11-12 retinas (when retina is still developing) is therefore more appropriate than using adult retina cells. In their rebuttal the authors outline why they selected this specific age in relation to retinopathy of prematurity. I would suggest adding some of this information to the appropriate part of the result section to make clear why this particular age has been selected.

In conclusion, the data in the manuscript are strong, provide new information, and likely to be of interest to the field. In my opinion, the authors have addressed all of the authors comments. However, there are a few small text changes that could be made to further clarify some of the points being addressed.

REVIEWERS' COMMENTS:

Reviewer #3 (Remarks to the Author):

I was asked to focus my review on the degree to which the authors have responded to the comments of reviewer 2.

1. The reviewer raises the very important point that the immortalised cells may differ in their behaviour and/or metabolism to primary Müller glial cells. In my opinion, the authors have been robust in their attempts to address this criticism. Most importantly, they have used primary (mouse) Müller glial cells to confirm their findings. They also have performed additional experiments using primary human brain cortical astrocytes and retinal explants.

2. The reviewer requested that throughout the manuscript the authors should refer to the MIO-M1 cell line when it is used (rather than referring to them as Müller glia). This has been done.

3. The reviewer quite correctly states that expression of a few markers is not sufficient to validate the identity of the MIO-M1 cells. Validation by the authors has been done using only a couple of markers. However, as noted above, the results have been confirmed using primary cells, demonstrating good functional equivalence between the immortalised cells and primary Müller glia cells. The authors also mention in their rebuttal that the cells have been characterised previously by others. However, I did not see any mention of this in the manuscript (apologies if I missed it). I therefore have a couple of suggestions for very minor text changes that would help to further address points 1-3 raised by the reviewer:

(i) In the abstract it would be helpful to clarify that both primary cells and a cell line have been used. For example, the 3rd sentence of the abstract could be modified along these lines: Using a stable isotope labelling technique in human RECs, mouse Müller glial cells and a human Müller glia cell line (M10-M1 cells),.....

We have added to the abstract the phrase as suggested by Reviewer 3:

Abstract

Although supplemental oxygen is required to promote survival of severely premature infants, hyperoxia is simultaneously harmful to premature developing tissues such as in the retina. Here we report the effect of hyperoxia on central carbon metabolism in primary mouse Müller glial cells and a human Müller glia cell line (M10-M1 cells). We found decreased flux from glycolysis entering the tricarboxylic acid cycle in Müller cells accompanied by increased glutamine consumption in response to hyperoxia. In hyperoxia, anaerobic catabolism of glutamine by Müller cells increased ammonia release two-fold. Hyperoxia induces glutamine-fueled

anaplerosis that reverses basal Müller cell metabolism from production to consumption of glutamine.

(ii) At the start of the results section (for example, after 1st sentence of paragraph 1 of the results), it would be useful to include a short statement explaining what M10-M1 cells are (an immortalised Müller glia cell line) and citing previous work demonstrating the “faithfulness” of these cells to Müller cells (for example the work by Astrid Limb mentioned in the rebuttal). The work done by the authors to characterised the cells themselves also could potentially be mentioned here.

We have added to the start of the Results section, “We first used MIO-M1, an immortalized cell line, to study the effect of hyperoxia on metabolism. MIO-M1 were isolated from human eye and have been reported to behave like primary Muller cells.” We have added the citation by Limb et al.

(iii) A the start of the Discussion it would be helpful to again clarify that data are from both primary Müller glia and M10-M1 cells (e.g. Our data obtained using both mouse Müller glial cells and M10-M1 cells (an immortalised human Müller glia-like cell line), demonstrate that....

At the start of the Discussion we have added the sentence, “Our data obtained using both primary mouse Müller glial cells and MIO-M1 (an immortalized human Müller glia-like cell line), demonstrate that Müller cells in hyperoxia have a higher flux through the glutaminolytic branch to the TCAC rather than from glycolysis.”

4. The reviewer questioned the use of dissociated retinal cells from P11-P12 mice, and states that adult retina should have been used instead. I agree with the authors that P11-P12 retina is the appropriate age to have been used in their experiments given the question being addressed. Retinopathy of prematurity is a problem that affects developing, not mature retinas. Using P11-12 retinas (when retina is still developing) is therefore more appropriate than using adult retina cells. In their rebuttal the authors outline why they selected this specific age in relation to retinopathy of prematurity. I would suggest adding some of this information to the appropriate part of the result section to make clear why this particular age has been selected.

We have added to the results, page 11, the text,, “Retinopathy of Prematurity is caused by oxygen supplementation necessary to resuscitating severely premature infants that unfortunately creates retinovascular growth attenuation and vasoobliteration that is the hallmark of phase 1, which subsequently leads to profound ischemia and abnormal angiogenesis in phase 2. Therefore, we chose specifically the hyperoxic phase 1 in the experimental correlate of ROP, the murine oxygen induced retinopathy model (OIR) ¹⁶, to test both primary Müller cells and retinal explants. Hyperoxic phase 1 in the mouse model of OIR is from P7 to P12, and to be consistent with the model we have only used cells or retinal explants from mice within the phase 1 of the model.”

In conclusion, the data in the manuscript are strong, provide new information, and likely to be of interest to the field. In my opinion, the authors have addressed all of the authors comments. However, there are a few small text changes that could be made to further clarify some of the points being addressed.

Thank you once again for your time and consideration.

Jonathan Sears